# STAR: Rethinking MoE Routing as Structure-Aware Subspace Learning

**Sumin Park** [1]    **Noseong Park** [1]

## Abstract

Mixture-of-Experts (MoE) scales model capacity efficiently by selectively routing inputs to a specialized subset of experts. However, input-expert specialization, the core motivation of MoE, critically depends on whether the router is actually aware of input structure. In practice, MoE routing is typically implemented as a shallow linear projection with limited awareness of input representation, which often leads to unstable routing. We propose **STAR**, a Structure-Aware Routing that rethinks MoE routing as a subspace learning problem by augmenting standard learnable routing with an evolving principal subspace that tracks dominant input structure via Generalized Hebbian Algorithm (GHA). By aligning routing decisions directly with input structure, STAR enables stable expert specialization. We evaluate STAR on controlled synthetic setup and large-scale language and vision tasks, where it consistently improves routing quality and downstream performance over strong MoE baselines. Moreover, optional test-time subspace updates further enhance routing robustness and generalization under input distribution shifts. Code is available at https://github.com/psmiz/STAR.

## 1. Introduction

The Mixture-of-Experts (MoE) (Jacobs et al., 1991; Jordan & Jacobs, 1993), is a classic design that substantially scales up the model capacity with minimal computation overheads. Recently, incorporating MoE into the deep neural networks has achieved remarkable successes (Eigen et al., 2014; Shazeer et al., 2017). Among the various variants of MoE, a sparsely gated MoE, namely SMoE, efficiently scales up Transformers (Lepikhin et al., 2020; Fedus et al., 2022), resulting in significant accuracy enhancements. Con-

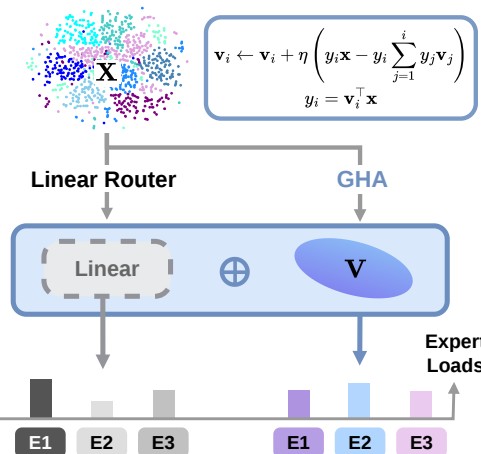

*Figure 1.* Overview of routing strategy of STAR.

sisting of multiple subnetworks (experts) that specialize in different tasks, MoE can benefit from a large pool of specialized knowledge with at modest computational cost by selective input gating to a subset of these experts. This scalable and flexible nature of MoE is particularly appealing for Large Language Models (LLMs), which struggle with massive model sizes and diverse training datasets. Recent LLMs actively incorporate MoE layers (Jiang et al., 2024; Dai et al., 2024; Qwen et al., 2025; Zhu et al., 2024; DeepSeek-AI et al., 2024) in their architecture.

The gating mechanism is the core component that governs the performance of MoE models, as it determines how each input is routed to a subset of experts. Despite its central role, current routing design remains overly simplistic, typically a shallow linear projection followed by softmax or sigmoid activation (Fedus et al., 2022; Jiang et al., 2024; Aghdam et al., 2024; Dai et al., 2024; Zhu et al., 2024; Qwen et al., 2025). Such a minimal design has limited capacity to capture the complex variation in the input distribution, which often results in unstable and imbalanced expert specialization (Shazeer et al., 2017).

To address this, prior work largely treats routing imbalance as the main target to be resolved. The most widely adopted approach adds an auxiliary regularizer to force balanced load allocation, such as the load-balancing loss (Shazeer et al., 2017) and the GShard loss (Lepikhin et al., 2020), or alternative routing paradigms such as expert-choice routing

[1]Korea Advanced Institute of Science and Technology (KAIST), Daejeon, Republic of Korea. Correspondence to: Noseong Park <noseong@kaist.ac.kr>.

*Proceedings of the 43rd International Conference on Machine Learning*, Seoul, South Korea. PMLR 306, 2026. Copyright 2026 by the author(s).

(Zhou et al., 2022; Wang et al., 2024a). However, while load balancing is essential for efficient expert utilization, it alone does not ensure that the routing decisions reflect the input variations (Wang et al., 2024b; Qiu et al., 2025). Fundamentally, the role of the gating mechanism is to be data-aware, discerning and responding to meaningful variations in the input so as to promote stable input–expert specialization. This points to a complementary, orthogonal axis that prior balancing-centric work leaves underexamined: whether the router is aware of input structure at all.

Motivated by this perspective, we propose **STAR** (Structure-Aware Routing), an input-aware routing framework that augments standard MoE routing with data-driven structural information learned in an unsupervised manner. Specifically, STAR incrementally learns a low-dimensional routing subspace that captures dominant directions of input variation and combines this with a task-supervised linear gate to construct the final routing logits. This design enables routing decisions to explicitly reflect underlying input structure while still guided by gradient signals optimized to downstream tasks. By targeting input-awareness as a complementary dimension to load balance, STAR improves expert specialization through structure-aware routing while remaining fully compatible with explicit balancing losses. Figure 1 shows a comprehensive overview of our approach.

To evaluate the effectiveness of structure-aware routing, we conduct experiments in both controlled synthetic settings and large-scale real-world benchmarks. We first adopt a synthetic sequential modeling task based on multinomial Hidden Markov Models (HMMs), following the GINC data generation framework (Xie et al., 2022). We then validate our approach on a wide range of language and vision benchmarks spanning pretraining, fine-tuning, and out-of-distribution (OOD) scenarios, where STAR consistently improves routing quality and downstream task performance over other MoE baselines.

## 2. Backgrounds

**Mixture of Experts** In Transformer-based models, a Mixture-of-Experts (MoE) layer, which consists of multiple expert networks $\{f_1, ..., f_K\}$ and a gating network $G$, i.e., router, replaces the feed-forward network (FFN) after the self-attention. The router, typically implemented as a shallow linear-softmax network, determines which subset of experts should process each input. Based on the design choice of gating functions, the MoE can be classified into two types, dense MoE that activates all experts weighted by gating coefficients (Pan et al., 2024; Dou et al., 2024) and sparse MoE (SMoE) that sparsely activates a few subset of experts, typically with top-k mechanism (Fedus et al., 2022; Dai et al., 2024). To efficiently scale up the model, SMoE has been widely preferred over dense variants for practical

use (Cai et al., 2025; Li et al., 2023). Throughout the paper, we will also focus on improving the sparse gating.

**Generalized Hebbian Algorithm** As a generalization of Oja's rule (Oja & Karhunen, 1985), which provides an incremental solution for estimating the first eigenvector of the data covariance, i.e., the first principal component (PC) of PCA, the Generalized Hebbian Algorithm (GHA) (Sanger, 1989), an unsupervised learning algorithm, incrementally estimates the top-$K$ principal components of a data distribution. Given input vectors $x \in \mathbb{R}^d$, GHA learns an orthonormal basis matrix $V \in \mathbb{R}^{K \times d}$ such that each row $v_i$ approximates the $i$-th principal direction of the input covariance matrix. At each iteration, GHA performs the following update for each component $i \in \{1, \ldots, K\}$:

$$v_i \leftarrow v_i + \eta \left( y_i x - y_i \sum_{j=1}^{i} y_j v_j \right), \quad y_i = v_i^\top x,$$

where $\eta$ is the learning rate controlling the step size of GHA updates. Iterative updates of $v_i$ following this rule enforces orthogonality among the learned components while aligning them with the dominant directions of input variance. Unlike standard PCA, GHA operates online and doesn't require explicit computation of the covariance matrix, making it suitable for streaming or mini-batch scenarios. Since PCA components correspond to the right singular vectors of the data matrix $X$ up to scale (i.e., the matrix $V$ in SVD of $X = U\Sigma V^\top$), GHA can be seen as an online approximation of the top-$K$ right singular vectors of $X$.

**Multinomial Hidden Markov Model** A Hidden Markov Model (HMM) (Baum & Petrie, 1966) is a generative probabilistic model for sequential data where each observation is emitted by a latent state, and these latent states evolve by a Markov process where each hidden state depends only on the previous state and emits an observation independently by emission probabilities. Formally, let $\{h_t\}_{t=1}^T$ be the hidden state sequence and $\{o_t\}_{t=1}^T$ the observations. The generative process is defined by an initial distribution $\pi = p(h_1)$, transition probabilities $\mathcal{H} = p(h_t|h_{t-1})$, and emission probabilities $\mathcal{E} = p(o_t|h_t)$. The joint probability of an entire sequence of hidden states and observations, parameterized by $\theta = \{\pi, \mathcal{H}, \mathcal{E}\}$, is given by:

$$p(o_{1:T}, h_{1:T} \mid \theta) = \pi(h_1) \prod_{t=2}^{T} p(h_t \mid h_{t-1}) \prod_{t=1}^{T} p(o_t \mid h_t)$$

A multinomial HMM refers to a HMM where the emission probabilities are modeled by a categorical (multinomial) distribution over a finite set of discrete symbols. Each hidden state $h_t \in \{1, \ldots, N\}$ is associated with a probability distribution over a finite set of observations $\mathcal{O} = \{1, \ldots, M\}$. The emission probabilities are defined by an emission matrix $\mathcal{E} \in \mathbb{R}^{N \times M}$, with each row specifying the likelihood

of emitting each symbol given a state. The model follows the standard HMM structure with an initial state distribution $\pi \in \mathbb{R}^N$ and a transition matrix $\mathcal{H} \in \mathbb{R}^{N \times N}$. This particular model is well suited for sequential data consisting of discrete symbols, such as natural language.

## 3. Proposed Method

---
**Algorithm 1** STAR
---

1: **Given:** input $X \in \mathbb{R}^{N \times d}$, trainable gating matrix $W_g \in \mathbb{R}^{K \times d}$, GHA basis $V \in \mathbb{R}^{K \times d}$, basis mixing matrix $R \in \mathbb{R}^{K \times K}$, interpolation coefficients $\alpha \in \mathbb{R}^K$, expert matrices $\{W_k\}_{k=1}^K$, iteration count per GHA update $m$.
2: Randomly initialize $W_g, V, R, \alpha, \{W_k\}_{k=1}^K$
3: **for** each input $x \in \mathbb{R}^d$ **do**
4:     GHA update with $m$ iterations per each forward pass:
5:     **for** $m$ iterations **do**
6:         **for** each $k$ in $\{1, ..., K\}$ **do**
7:             $v_k \leftarrow v_k + \eta y_k \left( x - \sum_{i=1}^k y_i v_i \right), y_k = v_k^\top x$
8:             $v_k \leftarrow v_k / \|v_k\|_2$
9:         **end for**
10:     **end for**
11:     Compute routing probabilities after basis mixing:

$$l_{\text{linear}} = xW_g^\top, \; l_{\text{GHA}} = xZ^\top, \; Z = RV$$
$$s = \text{Softmax}(\sigma(\alpha) \odot l_{\text{linear}} + (1 - \sigma(\alpha)) \odot l_{\text{GHA}})$$

14: **end for**

---

We propose **STAR** (Structure-Aware Routing), a routing framework that augments standard MoE gating with an evolving input subspace incrementally learned during training. The key idea is to make routing decisions reflect the dominant structure of the input, which we formalize as its principal subspace.

**Definition 3.1** (Principal input structure). Let $x \in \mathbb{R}^d$ be a hidden representation with a zero-mean assumption $\mathbb{E}[x] = 0$ and covariance $\Sigma_x := \mathbb{E}[xx^\top]$. The rank-$K$ *input structure* of $x$ is the principal subspace satisfying

$$S_K^* := \underset{P \in \mathbb{R}^{d \times K}, \, P^\top P = I}{\arg\min} \mathbb{E}\big\| x - PP^\top x \big\|_2^2, \quad (1)$$

whose columns are the top-$K$ eigenvectors of $\Sigma_x$, equivalently the directions of maximal projected variance of $x$.

STAR estimates this subspace online via the Generalized Hebbian Algorithm (GHA), maintaining an orthonormal basis $V \in \mathbb{R}^{K \times d}$ whose rows span $S_K^*$ and approximate the top-$K$ eigenvectors of $\Sigma_x$ (equivalently, the top-$K$ right singular vectors of $X$), without explicitly forming $\Sigma_x$.

On top of this orthonormal basis $V$, STAR introduces a learnable mixing matrix $R \in \mathbb{R}^{K \times K}$ that maps the principal directions into expert-specific routing vectors, giving

the structure-aware routing matrix $Z = RV$ whose rows serve as per-expert routing vectors. Without mixing (i.e., $R = I$), each routing vector of $Z$ corresponds to a single principal component and thus directly inherits the variance hierarchy of $V$. In this case, the experts aligned with the leading directions dominate routing, while those tied to low-variance directions are rarely selected, causing imbalanced utilization. Therefore, $R$ decouples the expert selection from this variance ordering of $V$ while keeping $Z$ grounded in the input structure spanned by $V$. We further analyze the balancing effect of basis mixing, comparing no mixing $R = I$, fixed random, and learnable $R$, in Section 4.3.

Given an input $x \in \mathbb{R}^d$, STAR forms combined routing logits from a task-supervised linear gate, $l_{\text{linear}} = xW_g^\top$ with trainable gating matrix $W_g \in \mathbb{R}^{K \times d}$, and from the structure-aware subspace, $l_{\text{GHA}} = xZ^\top$. The final routing scores $s \in \mathbb{R}^K$ interpolate the two via a learnable coefficient $\alpha \in \mathbb{R}^K$ as

$$s = \text{Softmax}(\sigma(\alpha) \odot l_{\text{linear}} + (1 - \sigma(\alpha)) \odot l_{\text{GHA}}),$$

where $\sigma(\cdot)$ is the elementwise sigmoid. Thus, STAR provides a unified routing framework that extends conventional MoE routing with explicit awareness of input structure. The full algorithmic details are summarized in Algorithm 1.

**Basis Construction** We apply $m$ iterative GHA updates at each forward pass, continually refining the principal directions in response to the current hidden features. The iteration number $m$, as a tunable hyperparameter, controls the trade-off between approximation quality and computation cost. To assess the quality of the GHA-driven basis, we compare it against SVD on hidden representations extracted from the Transformer encoder, as

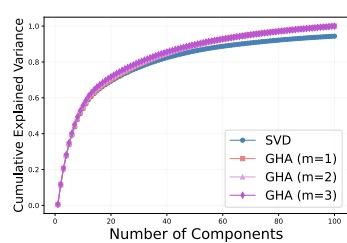

*Figure 2.* Comparison of cumulative explained variance.

shown in Figure 2. We measure the cumulative explained variance of the top 100 components with varying iteration counts $m \in \{1, 2, 3\}$ per GHA update. GHA closely tracks SVD even with small $m = 1$, demonstrating that GHA effectively approximates the principal subspace of the input distribution in an incremental manner. Further basis quality analysis under various experimental setup is provided in Appendix C.

## 4. Synthetic Experiments

We construct a synthetic task based on a structured language modeling scenario using Multinomial HMMs, following the GINC dataset generation procedure (Xie et al., 2022).

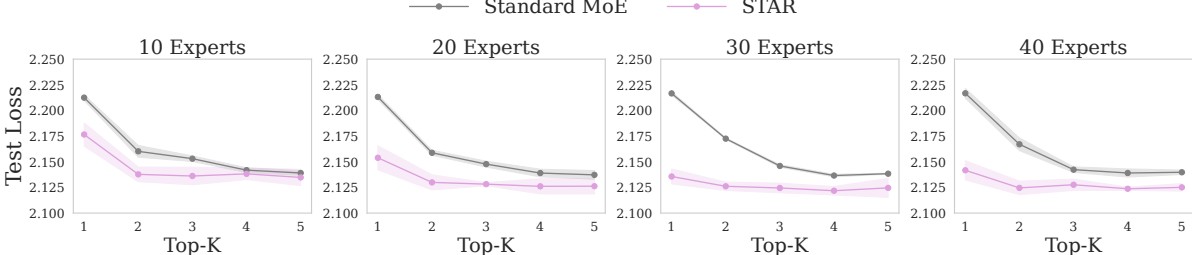

*Figure 3.* Performance comparison of standard MoE and STAR. Standard MoE: standard linear router + load balance regularizer. Test loss under varying numbers of experts and top-$k$. Shaded regions indicate standard deviation over three random seeds.

This provides the data with clear and interpretable structure, allowing us to rigorously isolate the impact of router design on expert specialization and routing stability.

### 4.1. Experimental Setup

**Base Layer** We define the base MoE architecture shared across all synthetic experiments. The model consists of two main components: a gating module $G$ and a set of experts $\{f_k\}_{k=1}^K$ where $W_k \in \mathbb{R}^{d' \times d}$ is the weight matrix of expert $k$. For expert selection, we adopt a top-$k$ selection mechanism throughout the paper, where $k$ is a predefined hyperparameter. Given the input $x \in \mathbb{R}^d$, the routing score for expert $k$ is computed as $s_k = \text{Softmax}_k(x^\top g_k)$, where $g_k \in \mathbb{R}^d$ is the $k$-th gating vector. The final output $\hat{y} \in \mathbb{R}^{d'}$ of the MoE is computed as $\hat{y} = \sum_{k \in \text{Topk}(\{s_j\}_{j=1}^K, k)} s_k \cdot f_k(x)$ where $f_k(x) = W_k x$.

**Standard MoE** We compare STAR against the most widely adopted gating design across recent MoE architectures (Fedus et al., 2022; Jiang et al., 2024; Aghdam et al., 2024; Dai et al., 2024; Zhu et al., 2024; Qwen et al., 2025), which we denote as Standard MoE. This corresponds to a standard linear projection with gating weight $W_g \in \mathbb{R}^{K \times d}$, where $g_k = \{W_g\}_k$ is the trained gating vector of the $k$-th expert, following common practice with a load-balancing regularizer (Shazeer et al., 2017).

**Data Generation** To model the synthetic language-like data, tokens are generated from latent entity-property states, temporally evolving by mixtures of HMMs, each carrying distinct context. Each hidden state is a tuple $h_t = (v_t, s_t)$, where $v_t \in \mathcal{V}$ and $s_t \in \mathcal{S}$ represents entity and property, respectively. A global memory matrix $M \in \mathcal{O}^{|\mathcal{V}| \times |\mathcal{S}|}$ maps each entity–property pair deterministically to a token in the vocabulary $\mathcal{O}$. The transitions are defined as:

$$s_{t+1} \sim P(s_{t+1} \mid s_t; \theta), \quad \theta \sim p(\theta),$$
$$v_{t+1} \sim 0.9 \cdot \delta(v_{t+1} = v_t) + 0.1 \cdot I(v_{t+1} \mid v_t),$$

where $\theta \in \Theta$ is a concept parameter sampled from a uniform mixture of HMMs, each governing distinct property

transition pattern. The $\delta(v_{t+1} = v_t)$ enforces entity persistence, yielding a sticky Markov process that keeps the same entity with probability 0.9 and switches uniformly with probability 0.1. At each step the emitted token is deterministic, $o_t = M[v_t, s_t]$ with $p(o_t \mid v_t, s_t) = 1$. Let $T$ be the sequence length, $n$ the number of training samples, and $t'$ the subsequence length. Each sample $(x_i, y_i)$ consists of input $x_i = [o_{t_i}, \ldots, o_{t_i+t'-1}]$ and target $y_i = o_{t_i+t'}$. Thus, each sample uses $t' + 1$ tokens, giving total length $T = n(t' + 1) + t'$. The full sequence is $[x_1, y_1, o_{\text{delim}}, \ldots, x_n, y_n, o_{\text{delim}}, x_{\text{test}}]$

**Model** We adopt a single-layer Transformer architecture whose final representation is augmented with a Mixture-of-Experts (MoE) layer. The model consists of a token embedding layer, sinusoidal positional encoding, and a Transformer encoder block. The MoE module is placed as the feed-forward network (FFN) head, replacing the standard MLP. It consists of $K$ expert networks, each implemented as a single linear layer followed by a SiLU activation, mapping from $\mathbb{R}^d$ to $\mathbb{R}^{d'}$. A gating network computes token-wise routing scores over experts, and the top-$k$ experts are selected for each token. A shared decoder layer maps the output back to the token vocabulary. The model is trained under standard next-token prediction paradigm.

### 4.2. Results

We configure the synthetic language modeling task with a vocabulary size of $|\mathcal{O}| = 50$, $|\mathcal{V}| = 15$ entities, $|\mathcal{S}| = 15$ properties, and context number $|\Theta| = 5$. After training the model, we measure the test loss across three random seeds to evaluate our proposed method against the Standard MoE baseline. Figure 3 presents the performance comparison across varying numbers of experts, $K \in \{10, 20, 30, 40\}$, and different top-$k$ selections, $k \in \{1, 2, 3, 4, 5\}$. STAR consistently achieves lower test loss than Standard MoE across all configurations and top-$k$ selections. The performance gap becomes more pronounced as the number of experts increases, indicating that STAR scales more robustly with model size and mitigates the degradation in routing quality commonly observed with larger expert pools.

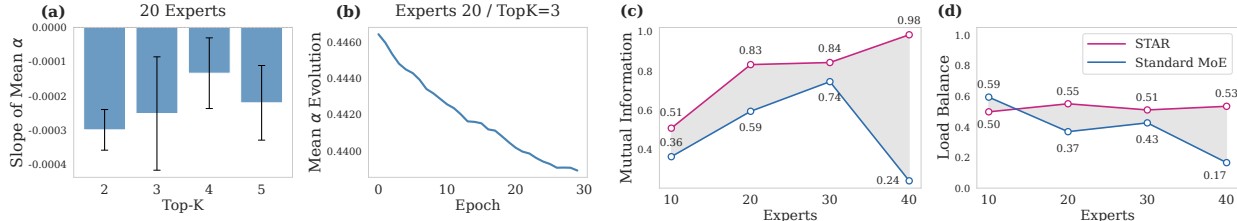

*Figure 4.* Comprehensive analysis on synthetic results. $(a)$: Average slope of the mean $\sigma(\alpha)$ across different Top-$k$ settings. $(b)$: Temporal evolution of mean $\sigma(\alpha)$ throughout training epochs. $(c)$: Routing specialization comparison measured by expert-property mutual information $I(e, s)$. $(d)$: Load balance comparison measured by normalized load balance $H_{\text{norm}}$.

**Interpolation Balance**    To investigate how $\alpha$ evolves to weight the GHA-driven gating against the learnable gating, we track the mean interpolation coefficient $\sigma(\alpha)$ during training. Recall that $\alpha$ controls the weighting between the two logits $l_{\text{linear}}$ and $l_{\text{GHA}}$. Figure 4-(a), (b) show the slope of mean $\sigma(\alpha)$ across different top-$k$ settings and its trajectory over epochs. We observe a consistent decrease in $\sigma(\alpha)$ throughout training, indicating a uniform shift toward greater reliance on the GHA-driven gating. This trend reflects the stabilization of hidden representations, which enables the model to increasingly favor structure-aware subspace gating over gradient-driven gating across all expert sizes and top-$k$ settings, confirming the robustness of this adaptation behavior. See Appendix F for full results.

**Specialization and Load Balance**    Further, to understand whether the router becomes actually aware of the underlying input structure, and whether such input-awareness translates into stronger input–expert specialization, we examine the correlation between routing decisions and the semantic components of the generated data. In our synthetic setup, the latent property $s_t$ is the primary source of token-level variation, making it the most informative target for assessing whether the gating mechanism responds meaningfully to changes in the input. Let $e_t \in \{1, \ldots, K\}$ denote the expert selected for token $x_t$ under top-1 routing scenario. We quantify structure-aligned specialization via the mutual information between expert assignments and latent properties,

$$I(e, s) = \sum_{e=1}^{K} \sum_{s \in \mathcal{S}} p(e, s) \log \frac{p(e, s)}{p(e)\, p(s)}, \qquad (2)$$

$p(e, s)$ is the empirical joint distribution of expert–property pairs and $p(e)$ and $p(s)$ are its marginals. Higher values of $I(e, s)$ indicate experts specialize along the true variation modes of the data rather than fragmenting arbitrarily.

In parallel, we evaluate how well each gating mechanism balances expert usage, since MoE performance mainly depends on preventing expert collapse. Let $p(e)$ denote the empirical fraction of tokens routed to expert $e$. We measure

load-uniformity using the normalized entropy

$$H_{\text{norm}} = -\frac{\sum_{e=1}^{K} p(e) \log p(e)}{\log K}, \qquad (3)$$

which lies in $[0, 1]$ and attains 1 when all experts are used uniformly. These two complementary metrics evaluate how well experts specialize ($I(e, s)$) and how evenly they are used ($H_{\text{norm}}$), giving a more holistic view of routing behavior as the expert pool scales.

Figure 4-(c), (d) report both quantities across different $K$ for STAR and the Standard MoE baseline. In panel (c), STAR consistently achieves higher mutual information $I(e, s)$, with the gap widening as $K$ increases: Standard MoE improves up to $K = 30$ but collapses at $K = 40$ ($I(e, s) = 0.24$), whereas STAR continues to rise, reaching 0.98. This indicates that STAR benefits from increased routing capacity more reliably and maintains stable specialization even at larger expert sizes. The panel (d) shows the normalized load entropy $H_{\text{norm}}$. Both methods are reasonably balanced for smaller expert sizes, but Standard MoE becomes increasingly imbalanced as $K$ grows, with $H_{\text{norm}}$ falling sharply from 0.43 at $K = 30$ to 0.17 at $K = 40$. In contrast, STAR maintains stable load balance across all configurations. Together, these results show that STAR's structure-aware routing scales to larger expert pools on both axes, sustaining strong specialization and, as a consequence, well-distributed expert utilization.

### 4.3. The Role of Mixing $R$: Decoupling the Variance Hierarchy

Here we analyze how the mixing matrix $R$ prevents GHA-driven routing from inheriting the variance hierarchy of the GHA basis. We first characterize the per-expert routing energy, which captures how strongly each expert is activated in terms of the variances along the input's principal directions (Lemma 4.1). This lets us contrast two analyzable cases: (i) no mixing ($R = I$), where the routing energy directly inherits the variance hierarchy and concentrates on the leading components, making expert collapse inevitable (Proposition 4.2); and (ii) random orthonormal mixing, where this spectral bias is removed and routing energy is equalized

in expectation across experts (Proposition 4.3). When $R$ is learnable, as in our default STAR setting, its dynamics are intractable and preclude a tight analytical characterization without restrictive assumptions. We instead evaluate the resulting routing energies empirically and show that a learnable $R$ likewise avoids the degenerate concentration of the $R = I$ case (Figure 5).

**Lemma 4.1** (Per-expert routing energy). *Let $x \in \mathbb{R}^d$ be a zero-mean random input with covariance $\Sigma_x := \mathbb{E}[xx^\top]$ and $V \in \mathbb{R}^{K \times d}$ have as its rows the top-$K$ orthonormal eigenvectors of $\Sigma_x$, with eigenvalues $\lambda_1 \geq \cdots \geq \lambda_K$. Define $y := Vx$, then*

$$\text{Cov}(y) = V\Sigma_x V^\top =: \Lambda = \text{diag}(\lambda_1, \ldots, \lambda_K).$$

*Let $R \in \mathbb{R}^{K \times K}$ and $Z := RV$ where $r_k^\top$ is the $k$-th row of $R$. For logits $\ell(x) := Zx$, define logit energy per expert $k$*

$$\mathcal{L}_k := \mathbb{E}_x[\ell_k(x)^2].$$

*Then we can express this as routing energy*

$$\mathcal{L}_k = \mathbb{E}_x[(r_k^\top y)^2] = r_k^\top \Lambda r_k. \tag{4}$$

*Equivalently, $\mathcal{L}_k = \sum_{i=1}^{K} \lambda_i r_{k,i}^2$ is a weighted sum of eigenvalues $\{\lambda_i\}$ with weights $r_{k,i}^2$. (Proof in Appendix B)*

**Proposition 4.2** (Routing energy with no $R$). *Under the setting of Lemma 4.1, we define an empirical diagonal covariance estimate*

$$\hat{\Lambda} = \text{diag}(\hat{\lambda}_1, \ldots, \hat{\lambda}_K)$$

*derived from samples $y = \widehat{V}x$ (e.g., $\hat{\lambda}_i = \widehat{\text{Var}}(y_i)$), where $\widehat{V}$ is an incremental GHA estimate of $V$. We use $\hat{\Lambda}$ as a consistent plug-in estimator for $\Lambda$. In the no mixing case, let $R = I$ (i.e., $Z = \widehat{V}$) where $r_k = e_k$ with $\|r_k\|_2 = 1$. Then the empirical routing energy of $k$-th expert is*

$$\hat{\mathcal{L}}_k := e_k^\top \hat{\Lambda} e_k = \hat{\lambda}_k \tag{5}$$

**Proposition 4.3** (Routing energy with random orthonormal $R$). *Let the identical setup as Proposition 4.2. Given $R$ is orthonormal such that $\mathbb{E}[r_k r_k^\top] = \frac{1}{K}I$, we obtain*

$$\mathbb{E}[\hat{\mathcal{L}}_k] = \mathbb{E}[r_k^\top \hat{\Lambda} r_k] = \frac{1}{K}\sum_{i=1}^{K} \hat{\lambda}_i, \quad \forall k. \tag{6}$$

Taken together, Propositions 4.2 and 4.3 show that without mixing ($R = I$), routing energy is intrinsically tied to the PCA spectrum of corresponding order and thus exhibits systematic imbalance across experts, whereas random orthonormal mixing removes this bias by equalizing routing energy in expectation across experts. This shows that the key

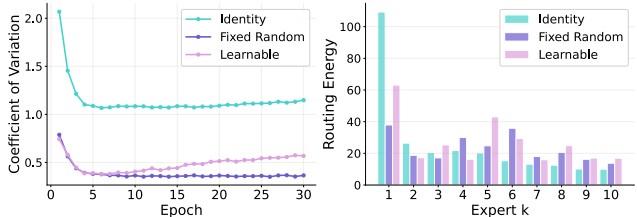

*Figure 5.* Routing energy analysis across varying $R$. Left: Coefficient of variation (CV) of per-expert routing energy over training, higher values indicate energy concentrated on fewer experts. Right: Per-expert routing energy distributions at the final epoch for $R = I$, fixed random orthonormal, and learnable $R$.

role of $R$ is to decouple expert selection from hierarchical variance ordering in input space.

Motivated by this contrast, we examine whether a learnable $R$ avoids the same collapse in practice. Figure 5 reports the empirical per-expert routing energy $\mathbb{E}[\ell_k(x)^2]$ in the synthetic setup with $K = 10$. In the left panel, direct PC routing ($R = I$) maintains a consistently high CV, reflecting strong expert dominance, whereas both fixed random and learnable $R$ substantially reduce it. The right panel shows the per-expert energy distribution at the final epoch, in which mixing with $R$ (fixed or learnable) yields a far more even spread than the highly skewed $R = I$ case.

## 5. Main Results on Real-Data Experiments

### 5.1. Zero-Shot Evaluation on Pretrained LLM-MoE

To assess whether STAR improves large-scale autoregressive modeling, we pretrain LLaMA-MoE backbones at two scales, with 182M and 469M active parameters (out of 777M and 2.6B total, respectively). Each model uses a standard LLaMA-style decoder (GQA, SwiGLU, RoPE, RMSNorm), with every feed-forward block replaced by an MoE layer of $E=8$ experts under Top-1 routing, matching the compute budget of the corresponding dense model. Following the pretraining setup of ReMoE (Wang et al., 2025), all models are trained from scratch on the Pile (Gao et al., 2020) for 30B tokens with sequence length 1024 and batch size 512. We compare STAR against three routing baselines: standard Top-1 MoE (Fedus et al., 2022), Expert-Choice (EC) routing (Zhou et al., 2022), and ReMoE (Wang et al., 2025).

In this large-scale pretraining setup, all methods, including STAR, are trained with the auxiliary load-balancing loss (Shazeer et al., 2017), as balanced expert utilization is important for stable large-scale MoE training. Applying it uniformly isolates the effect of structure-aware routing and confirms that STAR is complementary to explicit load balancing. We note that this is the only setting in which a balancing regularizer is applied and all other experiments train STAR without one. For STAR we set the GHA iteration count to $m=1$ for efficient training.

*Table 1.* Training curves and zero-shot performance of different MoE routing methods on LLaMA-MoE at the 182M and 469M scales. Left: training loss on the Pile. Right: zero-shot accuracy. All methods, including STAR, are trained with the auxiliary load-balancing loss. Best results in **bold** and second-best results underlined.

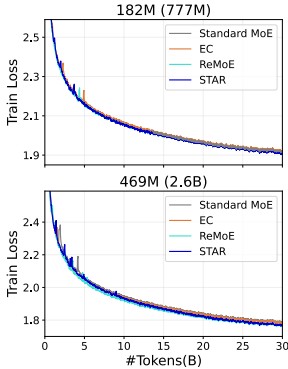

| Model | ARC-c | ARC-e | BoolQ | HellaSwag | LAMBADA | PIQA | RACE | Avg |
|---|---|---|---|---|---|---|---|---|
| *Active 182M / 777M* | | | | | | | | |
| Standard MoE | 24.15 | 40.78 | 59.88 | 33.56 | 33.98 | 64.25 | 27.94 | 40.65 |
| EC | 24.40 | 41.96 | 56.91 | 33.28 | 32.00 | 64.20 | 28.13 | 40.13 |
| ReMoE | 24.40 | **43.27** | 50.28 | 34.16 | 33.38 | 64.36 | **29.76** | 39.94 |
| **STAR** | **26.37** | 40.15 | **60.40** | **34.26** | **34.43** | **64.85** | 28.71 | **41.31** |
| *Active 469M / 2.6B* | | | | | | | | |
| Standard MoE | 24.23 | 44.61 | 52.75 | 38.98 | 40.15 | 67.08 | **31.00** | 42.69 |
| EC | 23.81 | 43.81 | **56.82** | 37.80 | 40.02 | 67.19 | 30.24 | 42.81 |
| ReMoE | **25.85** | **46.46** | 53.58 | **40.86** | 39.12 | 67.79 | 29.38 | 43.29 |
| **STAR** | 25.17 | 45.58 | 56.21 | 39.95 | **42.11** | **67.85** | 30.62 | **43.93** |

*Table 2.* Performance comparison on 5 subtasks on GLUE benchmark. All results averaged across 3 random seeds. [1]: Guo et al. (2025)

| Algorithms (K, k) | CoLA | MRPC | QNLI | MNLI | RTE | Average |
|---|---|---|---|---|---|---|
| DynMoE (9, 7.1)[1] | 65.17±0.26 | 90.64±0.26 | 92.59±0.08 | 86.37±0.13 | 73.41±1.96 | 81.64 |
| Cosine Router (8,1)[1] | 64.10±0.94 | 90.14±0.60 | 92.48±0.21 | 86.56±0.06 | 73.04±2.13 | 81.26 |
| Cosine Router (8,4)[1] | 64.94±0.62 | 89.74±0.99 | 92.52±0.12 | 86.57±0.28 | 75.09±1.84 | 81.77 |
| **STAR** (8,1) | 65.54±0.47 | 90.25±0.55 | 92.56±0.23 | 86.52±0.18 | 74.61±0.74 | 81.90 |
| **STAR** (8,4) | 66.62±0.65 | 89.68±0.20 | 92.61±0.10 | 86.74±0.11 | 75.57±0.68 | 82.24 |
| Cosine Router (16,1)[1] | 63.63±0.20 | 89.81±0.30 | 92.39±0.21 | 86.63±0.17 | 74.01±0.29 | 81.29 |
| Cosine Router (16,4)[1] | 64.12±1.42 | 89.74±0.40 | 92.65±0.09 | 86.59±0.16 | 75.33±0.95 | 81.69 |
| **STAR** (16,1) | 64.69±0.95 | 90.03±0.34 | 92.57±0.08 | 86.64±0.06 | 73.65±0.62 | 81.52 |
| **STAR** (16,4) | 65.74±1.11 | 90.20±0.37 | 92.61±0.05 | 86.59±0.07 | 75.41±0.68 | 82.11 |
| Ablations on STAR | | | | | | |
| No $R$ (8,4) | 64.57±0.82 | 90.12±0.50 | 92.54±0.04 | 86.38±0.06 | 72.56±1.77 | 81.23 |
| No Interpolation (8,4) | 66.02±1.32 | 89.56±0.31 | 92.39±0.17 | 86.51±0.12 | 73.53±2.45 | 81.60 |
| Random basis (8,4) | 65.80±0.69 | 90.20±0.67 | 92.54±0.25 | 75.52±15.85 | 73.89±2.47 | 79.59 |

Following pretraining, we evaluate all models in a purely zero-shot manner on seven commonsense and reading-comprehension tasks: ARC-c, ARC-e, BoolQ, HellaSwag, LAMBADA, PIQA, and RACE. Table 1 reports zero-shot accuracies and the corresponding training-loss curves. At both scales, STAR attains the highest average accuracy, improving over standard Top-1 MoE, EC, and ReMoE, and consistently tracks the lowest training loss throughout pretraining. While no single method dominates every individual task, STAR is the most consistent across tasks with either best or second-best results per task, indicating that structure-aware gating promotes effective expert specialization during large-scale pretraining.

### 5.2. Finetuning for MoE-Augmented BERT on GLUE

In this experiment, we follow the MoEfication setup (Zhang et al., 2022; Qiu et al., 2024) and finetune BERT-large (Devlin et al., 2019) models on the GLUE benchmark (Wang et al., 2019). Specifically, we evaluate on CoLA (Warstadt et al., 2019), QNLI (Wang et al., 2019), RTE (Bentivogli

et al., 2009), MNLI (Xu et al., 2020), and MRPC (Dolan & Brockett, 2005) under multiple MoE settings with varying $K$ and top-$k$ selections. We compare STAR against two representative MoE routing baselines. The cosine router (Li et al., 2023) replaces linear gating with cosine similarity between inputs and expert embeddings, improving routing stability through normalized similarity scoring. DynMoE (Guo et al., 2025) dynamically adjusts expert utilization by auto-tuning experts number during training, representing an adaptive MoE baseline. All results are averaged over three random seeds.

Table 2 shows that STAR achieves consistent improvements in average accuracy across all $(K, k)$ settings. At $E=8$, STAR outperforms both the cosine router and DynMoE at every top-$k$, with the best result at $(8, 4)$ reaching 82.24%, compared to 81.77% for the cosine router and 81.64% for DynMoE. The same trend holds at $E=16$, where STAR improves over the cosine router at matched $(K, k)$ (81.52 vs. 81.29 at $k=1$ and 82.11 vs. 81.69 at $k=4$), confirming that the gains persist as the expert pool grows.

*Table 3.* Comparison of ImageNet-C top-1 accuracy (%) on ViT-S/32 across 15 corruption types at severity levels 1, 3, and 5.

| | Noise | | | Blur | | | | Weather | | | | Digital | | | | Avg |
|---|---|---|---|---|---|---|---|---|---|---|---|---|---|---|---|---|
| Model | Gauss | Shot | Impulse | Defocus | Glass | Motion | Zoom | Snow | Frost | Fog | Bright | Contrast | Elastic | Pixelate | JPEG | |
| SEVERITY 1 | | | | | | | | | | | | | | | | |
| Standard MoE | 59.66 | 57.44 | 55.50 | 50.81 | 53.35 | 59.92 | 43.10 | 51.84 | 56.74 | **65.34** | 69.34 | **70.24** | 62.32 | 61.85 | 59.81 | 58.48 |
| **STAR** | **60.67** | **58.84** | **57.16** | 51.08 | 53.74 | 60.72 | 44.85 | 51.91 | 56.60 | 64.86 | 69.57 | 69.86 | **62.92** | 62.08 | 59.57 | 58.96 |
| **STAR (TTA)** | 60.24 | 58.56 | 56.24 | **51.23** | **54.20** | **60.87** | **45.55** | **51.94** | **57.21** | 65.24 | **69.80** | 69.98 | 62.78 | **62.13** | **60.08** | **59.07** |
| SEVERITY 3 | | | | | | | | | | | | | | | | |
| Standard MoE | 42.94 | 38.74 | 39.77 | **32.67** | 21.36 | 39.69 | 27.97 | 32.69 | 30.46 | 54.92 | 65.95 | **65.72** | 55.31 | 49.21 | 54.56 | 43.46 |
| **STAR** | 43.61 | **38.87** | 40.63 | 32.44 | 21.85 | **41.76** | 30.17 | 32.93 | 31.00 | 55.31 | **66.56** | 65.51 | 55.61 | 49.81 | **54.79** | 44.06 |
| **STAR (TTA)** | **43.76** | **38.87** | **40.65** | 32.46 | **21.86** | 41.74 | **30.19** | **32.97** | **31.43** | **55.54** | 66.55 | 65.50 | **55.93** | **49.83** | 54.77 | **44.14** |
| SEVERITY 5 | | | | | | | | | | | | | | | | |
| Standard MoE | **14.85** | **13.57** | **12.54** | 15.47 | 10.17 | 12.54 | 17.27 | 14.49 | 23.66 | 35.20 | 56.11 | 34.08 | 20.17 | 19.57 | 39.82 | 22.63 |
| **STAR** | 14.80 | 12.91 | 12.46 | 14.98 | 10.91 | **19.22** | **18.59** | 16.85 | 23.74 | 36.52 | 57.95 | **36.54** | 21.05 | 21.36 | 40.28 | 23.88 |
| **STAR (TTA)** | 14.80 | 13.33 | 12.49 | **16.09** | **10.94** | **19.22** | 18.55 | **16.94** | **24.72** | **37.78** | **57.96** | 36.52 | **21.10** | **21.80** | **40.57** | **24.19** |

**Ablations** We study the contributions of STAR's key components by ablating the mixing matrix $R$, the interpolation with the learnable gating matrix, and the use of an evolving subspace. As shown in Table 2, removing the mixing matrix $R$ and directly using the GHA basis as gating vectors consistently degrades performance, indicating that basis mixing is essential for effective routing. Eliminating interpolation and relying solely on the unsupervised basis also reduces accuracy, reflecting the importance of task-dependent signals provided by the gradient-guided gating. Finally, replacing the evolving GHA subspace with a fixed random orthonormal basis causes the largest drop (79.59%) and high instability (e.g., MNLI $75.52 \pm 15.85$), showing that STAR's gains arise from data-driven, incremental subspace learning. Together, these results confirm that all three components are necessary for STAR to achieve its full capacity.

### 5.3. Test Time Adaptation

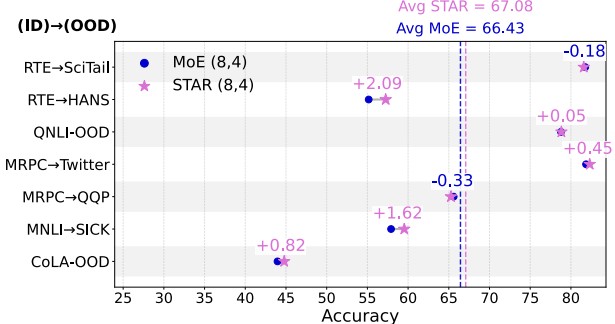

*Figure 6.* **OOD Performance comparison on GLUE-X.** The plot shows per-task accuracy differences between MoE and STAR, along with average improvements.

**GLUE-X for Language OOD Generalization** So far, all experiments have been conducted with GHA updates disabled at test time, leaving the gating basis fixed after training. In this section, we investigate the scenario where the unsupervised GHA updates are enabled during inference, allowing the router to adapt its basis to the test distribution in an online and unsupervised manner. Such adaptation is

particularly important in OOD scenarios, where test data deviates from training and a fixed gating basis may fail to generalize on the new structure.

We evaluate STAR on GLUE-X, which extends the GLUE benchmark with corresponding OOD test sets for each task. Models are finetuned on the in-distribution (ID) task and then evaluated on its OOD variant. All results are averaged over three random seeds. As shown in Figure 6, STAR achieves higher test accuracy than the MoE baseline with cosine router, same architecture with MoE in Table 2, on most OOD datasets (5 out of 7) with average improvement 0.65%p. Without any further parameter tuning, STAR reliably improves over the MoE baseline, indicating that enabling test-time adaptation through structure-aware gating provides more robust handling of distribution shifts.

**ImageNet-C for Vision OOD** We evaluate the TTA capability of STAR on ImageNet-C, a benchmark that applies 15 common image corruption types at multiple severity levels to ImageNet images (Hendrycks & Dietterich, 2019). We employ a ViT-S/32 (Dosovitskiy et al., 2021) model pretrained on ImageNet-1k (Dosovitskiy et al., 2021; Touvron et al., 2021) and apply MoEfication following the integration strategy of GMoE (Li et al., 2023), replacing the feedforward blocks with MoE layers. Each MoE-augmented model is then finetuned on 20k randomly sampled images from ImageNet-1k under its corresponding routing method. After finetuning, evaluation is performed on ImageNet-C as an out-of-distribution (OOD) test set. For STAR (TTA), we enable unsupervised GHA updates during inference so that the gating basis can adapt online to corrupted test inputs.

The results in Table 3 shows that across all three corruption severities, STAR improves over the Standard MoE baseline, and enabling test-time adaptation yields further gains. At modest corruption levels (severity 1 and 3), default STAR consistently improves over Standard MoE, and STAR (TTA) achieves the highest average accuracies at both severities. Under the most challenging corruption level (severity 5), STAR yields the largest improvement over Standard MoE,

and STAR (TTA) achieves the highest overall accuracy (24.19%). While TTA consistently provides additional gains across all severities, the improvements from STAR alone already indicate that its structure-aware routing contributes meaningful OOD resilience even without test-time updates.

## 6. Conclusion

Building on the view that effective expert specialization requires the router to recognize the dominant structure of the input, STAR augments the learnable gate with an evolving principal subspace estimated online via GHA. Across synthetic, large-scale pretraining, and fine-tuning settings, STAR consistently improves the downstream performance over standard MoE and recent routing baselines. In addition, STAR subspace can be optionally updated at test time to further reinforce robustness under distribution shift. A natural concern is whether the iterative GHA updates introduce computational overhead. We address this analytically and empirically in Appendix A, in which under typical MoE configurations, the overhead is negligible relative to the dominant expert computation. Overall, STAR offers a simple yet effective step toward structure-aware MoE routing that improves both specialization and generalization.

## Acknowledgements

This work was partly supported by the Institute for Information & Communications Technology Planning & Evaluation (IITP) grants funded by the Korean government (MSIT) (No. RS-2026-25526850, High-Efficiency Neural Networks for Artificial General Intelligence, 33%; No.2022-0-00857, Development of Financial and Economic Digital Twin Platform based on AI and Data, 33%; No. RS-2025-25442149, LG AI STAR Talent Development Program for Leading Large-Scale Generative AI Models in the Physical AI Domain, 1%), and Samsung Research Funding & Incubation Center of Samsung Electronics under Project Number SRFC-IT2402-08, 33%.

## Impact Statement

This work reframes the routing problem in Mixture-of-Experts (MoE) models, showing that effective expert specialization depends on the router's ability to recognize and respond to structure in the input rather than on the simplistic linear gating used in current designs. By making routing explicitly structure-aware, STAR can help large MoE models use their expert capacity more effectively and remain robust under distribution shift, which is increasingly important as MoE architectures scale. More broadly, the perspective we develop, treating routing as online subspace learning, suggests a principled, data-driven direction for designing and interpreting MoE routing mechanisms. We do not foresee negative societal impacts specific to this work beyond those already associated with large-scale models.

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

# A. Computation Analysis

In this section, we analyze the computational complexity of STAR in comparison to standard MoE gating mechanisms. We break down the cost of each stage, GHA basis updates, subspace mixing, and routing. The goal is to clarify the additional overhead introduced by GHA updates and demonstrate that it remains moderate relative to the overall cost of MoE forward and backward passes. Consider a mini-batch $X \in \mathbb{R}^{B \times d}$, hidden size $d$, number of experts (basis components) $K$, routing fan-out $k$ with top-$k$ selection, expert's width $d'$, and $m$ GHA iterations per pass.

**Standard Linear Gate.**

- Batchwise computation of routing logits: $XW_g^\top$ costs $\mathcal{O}(BKd)$.

- The $k$ selected expert MLPs dominate with $\mathcal{O}(Bkdd')$.

Thus the baseline batch cost is $\mathcal{O}(BKd) + \mathcal{O}(Bkdd')$.

**STAR.**

- GHA updates: Using the cached projection $Y = XV^\top$ and prefix cumsums, each iteration updates all $K$ rows in $\mathcal{O}(BKd)$ with $m$ iterations give $\mathcal{O}(mBKd)$.

- Basis mixing: We form $Z = RV$ once per pass, which results in $\mathcal{O}(K^2d)$.

- Extra logits: Given $Z$, computing the additional logits $l_{\text{GHA}} = XZ^\top$ costs $\mathcal{O}(BKd)$, and interpolation/softmax adds $\mathcal{O}(BK)$.

All other operations (softmax/top-$k$, expert MLPs) are unchanged from the baseline. Therefore, the batch cost of STAR is

$$\underbrace{\mathcal{O}\big(B((m+2)Kd)\big)}_{\text{GHA updates + extra logits}} + \underbrace{\mathcal{O}(K^2d)}_{\text{mixing, once per pass}} + \underbrace{\mathcal{O}(Bkdd')}_{\text{experts}}.$$

Hence, the incremental overhead over the baseline is

$$\mathcal{O}((m+1)BKd) + \mathcal{O}(K^2d).$$

Under the typical STAR setup where the ratio of expert number ($K$) and model dimension ($d$) is around 1–4% (Fedus et al., 2022; Jiang et al., 2024; Aghdam et al., 2024; Dai et al., 2024; Zhu et al., 2024; Qwen et al., 2025), we have $K \ll \min(d, d')$. In practice, we further adopt small number of iterations $m \in \{1, 3\}$. Under these practical conditions,

$$\mathcal{O}((m+1)BKd) + \mathcal{O}(K^2d) \ll \mathcal{O}(Bkdd').$$

Therefore, the additional computation introduced by STAR is negligible compared to the dominant expert MLP computation in the base MoE architecture, and is effectively equivalent to increasing the expert width $d'$ by a small constant factor, $m + 1$.

**Empirical Analysis of Runtime and Memory.** Table 4 reports runtime and memory usage statistics across different MoE models under $K = 8$ and top-$k$=4 setting. For training, we measure (i) micro-step time, the latency of a single forward/backward micro-batch, (ii) optimization-step time, the latency per parameter update including gradient accumulation, and (iii) peak GPU memory usage during training. For inference, we report (i) latency per example, averaged over multiple runs after warmup, and (ii) peak GPU memory usage during inference.

We observe that STAR achieves nearly identical or even smaller training cost to the MoE with Standard MoE and cosine router, despite performing three iterative GHA updates per forward pass. DynMoE takes up around 4 times higher time to run a single step run, attributed to the explicit tracking of token routing frequency. This is because the baseline router itself requires additional computation for regularizing the similarity matrix or load balance, while GHA updates scale linearly with $K$ and $d$ without balancing term, resulting in negligible overhead compared to dominant MoE operations. Consequently, STAR provides improved routing performance without incurring severe extra cost during training. At inference time, when

test-time GHA updates are disabled, STAR incurs only minimal latency and memory overhead by reusing the fixed gating basis derived during training, effectively matching the efficiency of the baseline. If test-time GHA updates are enabled, inference latency increases due to additional online updates, but this is optional and provides robustness under distribution shift. Overall, STAR offers a favorable trade-off, introducing no extra burden in training while supporting flexible test-time adaptation.

*Table 4.* Runtime and memory comparison. Times are mean±std (ms). Peak memory is in GiB.

| | **Inference** | | **Training (per step)** | | |
|---|---|---|---|---|---|
| **Method** | Latency ↓ | Peak Mem ↓ | Micro-step ↓ | Opt-step ↓ | Peak Mem ↓ |
| DynMoE | 17.2±9.1 | 5.2 | 428.4±168.7 | 80.2±11.9 | 11.4 |
| Standard MoE | 11.0±14.5 | 4.8 | 130.7±90.5 | 43.7±2.3 | 7.9 |
| MoE (cosine router) | 12.0±16.0 | 4.8 | 134.7±100.0 | 43.9±2.7 | 8.0 |
| STAR (m = 3) | 10.3±5.5 | 4.7 | 131.1±94.2 | 43.7±2.8 | 7.9 |
| STAR (TTA, m = 1) | 10.4±6.6 | 4.8 | 127.1±92.4 | 43.6±2.8 | 8.0 |
| STAR (TTA, m = 3) | 16.4±7.6 | 4.8 | - | - | - |

# B. Theoretical Derivations

**Lemma 4.1** (Per-expert routing energy). *Let $x \in \mathbb{R}^d$ be a zero-mean random input with covariance $\Sigma_x := \mathbb{E}[xx^\top]$ and $V \in \mathbb{R}^{K \times d}$ have as its rows the top-$K$ orthonormal eigenvectors of $\Sigma_x$, with eigenvalues $\lambda_1 \geq \cdots \geq \lambda_K$. Define $y := Vx$, then*

$$\mathrm{Cov}(y) = V\Sigma_x V^\top =: \Lambda = \mathrm{diag}(\lambda_1, \ldots, \lambda_K).$$

*Let $R \in \mathbb{R}^{K \times K}$ and $Z := RV$ where $r_k^\top$ is the $k$-th row of $R$. For logits $\ell(x) := Zx$, define logit energy per expert $k$*

$$\mathcal{L}_k := \mathbb{E}_x\big[\ell_k(x)^2\big].$$

*Then we can express this as routing energy*

$$\mathcal{L}_k = \mathbb{E}_x\big[(r_k^\top y)^2\big] = r_k^\top \Lambda r_k. \tag{4}$$

*Equivalently, $\mathcal{L}_k = \sum_{i=1}^K \lambda_i\, r_{k,i}^2$ is a weighted sum of eigenvalues $\{\lambda_i\}$ with weights $r_{k,i}^2$. (Proof in Appendix B)*

*Proof.* By definition $Z := RV$ and the $k$-th row of $R$ is $r_k^\top$. Hence the $k$-th routing logit satisfies

$$\ell_k(x) = z_k^\top x = (r_k^\top V)x = r_k^\top(Vx) = r_k^\top y.$$

Since $x$ is assumed to be zero-mean and $y = Vx$ is a linear transform with deterministic $V$, we have $\mathbb{E}[y] = 0$ due to the linearity of expectation. Therefore,

$$\mathcal{L}_k := \mathbb{E}\big[\ell_k(x)^2\big] = \mathbb{E}\big[(r_k^\top y)^2\big] = \mathbb{E}\big[r_k^\top yy^\top r_k\big]. \tag{7}$$

Taking expectation over $x$ (hence over $y$), we can pull it out $r_k$

$$\mathbb{E}\big[r_k^\top yy^\top r_k\big] = r_k^\top \mathbb{E}[yy^\top] r_k.$$

Given $y := Vx$ with zero-mean $x$, let $\Sigma_x := \mathrm{Cov}(x) = \mathbb{E}[xx^\top]$. Then

$$\mathrm{Cov}(y) = \mathbb{E}\big[(y - \mathbb{E}[y])(y - \mathbb{E}[y])^\top\big] = \mathbb{E}[yy^\top] = \mathbb{E}[Vxx^\top V^\top] = V\Sigma_x V^\top.$$

Given $V$ is orthonormal and its rows form an eigenbasis of $\Sigma_x$, then

$$V\Sigma_x V^\top = \Lambda,$$

where $\Lambda$ is diagonal. Hence

$$\mathrm{Cov}(y_i, y_j) = \lambda_i \text{ if } i = j, \ 0 \text{ for } i \neq j.$$

Substituting this back to Eq. (7) yields

$$\mathcal{L}_k = r_k^\top \Lambda r_k.$$

Finally, since $\Lambda = \mathrm{diag}(\lambda_1, \ldots, \lambda_K)$ is diagonal, we can expand the quadratic form

$$r_k^\top \Lambda r_k = \sum_{i=1}^{K} \sum_{j=1}^{K} r_{k,i} \Lambda_{ij} r_{k,j} = \sum_{i=1}^{K} r_{k,i} \Lambda_{ii} r_{k,i} = \sum_{i=1}^{K} \lambda_i r_{k,i}^2.$$

$\square$

**Proposition 4.2** (Routing energy with no $R$). *Under the setting of Lemma 4.1, we define an empirical diagonal covariance estimate*

$$\hat{\Lambda} = \mathrm{diag}(\hat{\lambda}_1, \ldots, \hat{\lambda}_K)$$

*derived from samples $y = \widehat{V} x$ (e.g., $\hat{\lambda}_i = \widehat{\mathrm{Var}}(y_i)$), where $\widehat{V}$ is an incremental GHA estimate of $V$. We use $\hat{\Lambda}$ as a consistent plug-in estimator for $\Lambda$. In the no mixing case, let $R = I$ (i.e., $Z = \widehat{V}$) where $r_k = e_k$ with $\|r_k\|_2 = 1$. Then the empirical routing energy of $k$-th expert is*

$$\hat{\mathcal{L}}_k := e_k^\top \hat{\Lambda} e_k = \hat{\lambda}_k \tag{5}$$

*Proof.* We prove each case under the setting of Lemma 4.1. Throughout, we use the empirical diagonal covariance $\hat{\Lambda} = \mathrm{diag}(\hat{\lambda}_1, \ldots, \hat{\lambda}_K)$ computed from samples $y = \hat{V} x$ as a consistent plug-in estimator of $\Lambda$.

If $R = I$, then $Z = R\widehat{V} = \widehat{V}$, and hence $\ell(x) = Zx = \widehat{V} x = y$. In particular, $\ell_k(x) = y_k$. By the assumption $\mathrm{Cov}(y) = \hat{\Lambda} = \mathrm{diag}(\hat{\lambda}_1, \ldots, \hat{\lambda}_K)$ and $\mathbb{E}[y] = 0$, we have

$$\mathcal{L}_k = \mathbb{E}\big[\ell_k(x)^2\big] = \mathbb{E}[y_k^2] = \widehat{\mathrm{Var}}(y_k) = \hat{\lambda}_k,$$

Moreover,

$$\frac{\max_k \mathcal{L}_k}{\frac{1}{K} \sum_{j=1}^{K} \mathcal{L}_j} = \frac{\max_k \hat{\lambda}_k}{\frac{1}{K} \sum_{j=1}^{K} \hat{\lambda}_j} = \frac{\hat{\lambda}_1}{\frac{1}{K} \sum_{j=1}^{K} \hat{\lambda}_j},$$

where we used the convention $\hat{\lambda}_1 \geq \cdots \geq \hat{\lambda}_K$. $\square$

**Proposition 4.3** (Routing energy with random orthonormal $R$). *Let the identical setup as Proposition 4.2. Given $R$ is orthonormal such that $\mathbb{E}[r_k r_k^\top] = \frac{1}{K} I$, we obtain*

$$\mathbb{E}[\hat{\mathcal{L}}_k] = \mathbb{E}[r_k^\top \hat{\Lambda} r_k] = \frac{1}{K} \sum_{i=1}^{K} \hat{\lambda}_i, \quad \forall k. \tag{6}$$

*Proof.* By Lemma 3.1,

$$\mathcal{L}_k = r_k^\top \hat{\Lambda} r_k.$$

Taking expectation over the random orthonormal $R$ and using $\mathbb{E}[r_k r_k^\top] = \frac{1}{K} I$, we obtain

$$\mathbb{E}[\mathcal{L}_k] = \mathbb{E}\Big[r_k^\top \hat{\Lambda} r_k\Big] = \mathbb{E}\Big[\mathrm{tr}(\hat{\Lambda} r_k r_k^\top)\Big] = \mathrm{tr}\Big(\hat{\Lambda}\, \mathbb{E}[r_k r_k^\top]\Big) = \mathrm{tr}\Big(\hat{\Lambda} \cdot \frac{1}{K} I\Big) = \frac{1}{K} \mathrm{tr}(\hat{\Lambda}) = \frac{1}{K} \sum_{i=1}^{K} \hat{\lambda}_i.$$

$\square$

## C. Basis Quality Analysis with Varying GHA Iterations

In this section, we analyze how the number of GHA update iterations per forward pass ($m$) affects the quality of the constructed basis. Since GHA operates incrementally and approximates the principal components through iterative updates, the choice of $m$ governs the trade-off between computational cost and approximation fidelity. We compare the cumulative explained variance of GHA-derived basis to that of full-batch SVD across three representative settings: (i) hidden states from the trained encoder of Transformer used in the second synthetic task, (ii) ResNet-18 features on CIFAR-100, and (iii)

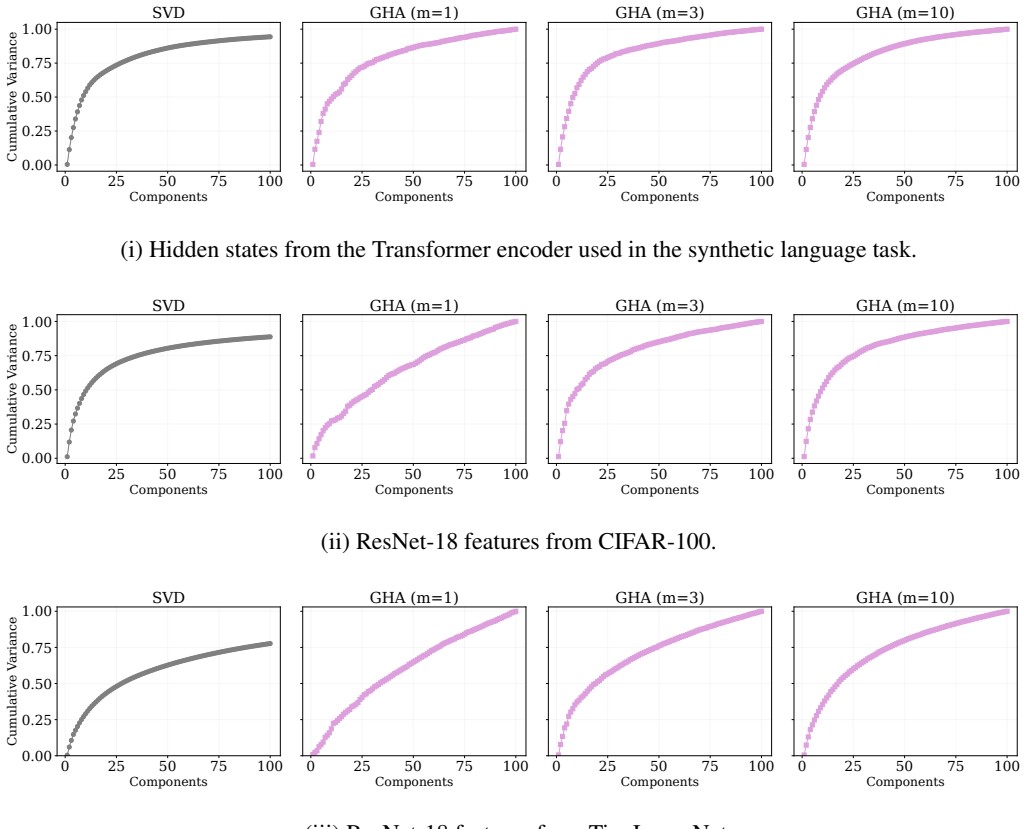

(i) Hidden states from the Transformer encoder used in the synthetic language task.

(ii) ResNet-18 features from CIFAR-100.

(iii) ResNet-18 features from TinyImageNet.

*Figure 7.* **Effect of GHA iteration number $m$ on basis approximation quality.** Cumulative explained variance of the top 100 components extracted using GHA with varying $m \in \{1, 3, 10\}$, compared to full-batch SVD. Results are shown for three settings: (i) Transformer hidden states on synthetic language datasets, (ii) ResNet-18 features on CIFAR-100, and (iii) ResNet-18 features on TinyImageNet. Higher $m$ improves approximation fidelity to SVD.

ResNet-18 features on TinyImageNet. In each case, we vary $m \in \{1, 3, 10\}$ and evaluate the variance captured by the top 100 components.

Figure 7 shows that as the number of GHA iterations increases, the cumulative explained variance curves increasingly align with those obtained from SVD. This confirms that higher $m$ enables GHA to better approximate the principal subspace. Overall, GHA provides a viable online alternative to SVD, with approximation quality controllable via $m$. Note that GHA fits a fixed number of components $K$, and the cumulative variance curve is normalized over these $K$ dimensions. In contrast, SVD operates over the full rank of the input data (i.e., $d$), and reports the ratio of total variance explained up to the top-$K$ components. As a result, GHA curves always end at 1.0 by construction, while the corresponding SVD curves may saturate earlier or lower depending on how much of the total variance is captured within the top-$K$ singular directions.

## D. Ablation Studies

We conduct ablation studies to analyze the sensitivity of STAR to its key design choices. In particular, we focus on two key aspects of STAR. First, we examine the effect of reducing the number of GHA updates $m$ performed at each training step. Since the iterative updates are designed to incrementally refine the principal subspace, varying $m$ allows us to understand how much adaptation is required to achieve stable and accurate routing. Second, we analyze the role of the GHA-driven basis itself in capturing the input structure and promoting expert specialization. To this end, we replace the learned principal basis with a fixed set of orthogonal random vectors while keeping the interpolation with the standard learnable gating matrix unchanged. This comparison disentangles the benefit of true data-driven subspace learning from simply mixing in an auxiliary random basis.

*Table 5.* Ablation study on the number of GHA iterations ($m$) in GLUE benchmark. Results are reported as Avg±Std over three random seeds.

| $m$ | $(K, k)$ | CoLA | MRPC | QNLI | MNLI | RTE | Average |
|---|---|---|---|---|---|---|---|
| 1 | (8, 1) | 64.68±0.83 | 90.52±0.15 | 92.40±0.18 | 86.47±0.03 | 74.49±0.45 | 81.71 |
|  | (8, 2) | 66.49±0.47 | 89.85±1.12 | 92.58±0.06 | 86.66±0.13 | 74.85±1.96 | 82.10 |
|  | (8, 4) | 65.36±1.17 | 90.10±0.35 | 92.40±0.15 | 86.48±0.14 | 75.33±1.19 | 81.94 |
| 2 | (8, 1) | 66.25±0.96 | 90.47±1.28 | 92.46±0.20 | 86.65±0.21 | 74.13±0.85 | 81.99 |
|  | (8, 2) | 65.93±2.19 | 90.03±0.32 | 92.59±0.18 | 86.76±0.08 | 74.24±0.34 | 81.91 |
|  | (8, 4) | 65.03±0.76 | 89.62±0.96 | 92.48±0.08 | 86.55±0.24 | 75.10±0.29 | 81.76 |
| 3 | (8,1) | 65.54±0.47 | 90.25±0.55 | 92.56±0.23 | 86.52±0.18 | 74.61±0.74 | 81.90 |
|  | (8,2) | 65.81±0.80 | 90.03±0.32 | 92.52±0.15 | 86.63±0.08 | 75.57±0.61 | 82.11 |
|  | (8,4) | 66.62±0.65 | 89.68±0.20 | 92.61±0.10 | 86.74±0.11 | 75.57±0.68 | 82.24 |

## D.1. Performance with Smaller GHA Updates

Table 5 reports results with smaller numbers of GHA iterations $m \in \{1, 2, 3\}$. While all settings consistently outperform the baseline MoE and DynMoE models, the choice of $m = 3$ yields the most consistent and superior performance across tasks. Smaller values of $m$ (e.g., $m = 1$ or $m = 2$) still provide competitive results, confirming that even limited iterative updates are sufficient to capture structural patterns, but additional updates stabilize training and improve generalization.

# E. Experimental Setup

We present the experimental setup used to evaluate STAR across language and vision tasks. This section first describes the datasets, covering the GLUE benchmark for language understanding and the DomainBed benchmark for domain generalization, and then outlines the training configurations and hyperparameters adopted for fair comparison with existing MoE baselines.

## E.1. Datasets

**Commonsense Reasoning.** We evaluate on zero-shot commonsense reasoning benchmark and reading-comprehension benchmark (Lai et al., 2017). For multiple-choice tasks, we report task accuracy on PIQA (Bisk et al., 2019), HellaSwag (Zellers et al., 2019), ARC-Easy, ARC-Challenge (Clark et al., 2018), LAMBADA (Paperno et al., 2016), and BoolQ (Clark et al., 2019).

**GLUE Benchmark (Language Tasks).** For language modeling, we follow the MoEfication setup (Zhang et al., 2022) and finetune BERT-large (Devlin et al., 2019) on the GLUE benchmark (Wang et al., 2019). We evaluate on five representative GLUE subtasks: CoLA (Warstadt et al., 2019) (linguistic acceptability), MRPC (Dolan & Brockett, 2005) (paraphrase detection), QNLI (Wang et al., 2019) (question-answer entailment), MNLI (Xu et al., 2020) (natural language inference), and RTE (Bentivogli et al., 2009) (textual entailment). These datasets jointly cover grammaticality, semantic similarity, and entailment, providing a comprehensive testbed for expert specialization in language understanding.

**GLUE-X OOD Benchmark.** To evaluate robustness under distribution shift, we additionally consider GLUE-X (Yang et al., 2023), an extension of GLUE that augments each in-distribution (ID) task with corresponding out-of-distribution (OOD) test sets drawn from different domains. GLUE-X provides 15 OOD datasets spanning eight GLUE tasks, enabling systematic assessment of cross-domain generalization. Importantly, models are trained only on the standard GLUE training sets and then directly evaluated on unseen OOD datasets, without exposure to target-domain data. In our experiments, we follow the GLUE-X protocol and evaluate STAR on selected OOD datasets corresponding to the five GLUE subtasks used in our ID experiments (CoLA, MRPC, QNLI, MNLI, and RTE). Specifically, we adopt CoLA → CoLA-OOD, MRPC → QQP / Twitter, QNLI → QNLI-OOD, MNLI → SICK, and RTE → SciTail / HANS, as shown in Figure 6.

**ImageNet-1k and ImageNet-C.** ImageNet-1k is a large-scale visual classification benchmark containing 1,000 object categories and 1.28M training images, serving as the pretraining source for our ViT-S/32 backbone. For robustness evaluation, we adopt ImageNet-C, which applies 15 corruption types spanning noise, blur, weather, and digital distortions, each at

severity levels 1–5. These corruptions simulate realistic distribution shifts and enable systematic assessment of OOD robustness. Following the standard protocol, models are trained on clean ImageNet-1k and evaluated directly on ImageNet-C without exposure to corrupted images during training.

## E.2. Hyperparameters and Configuration

Here we document the full training and evaluation configurations used across all experimental settings in this work.

*Table 6.* Pretraining configuration for the 182M and 469M LLaMA-MoE models.

| Category | Hyperparameter | 182M | 469M |
|---|---|---|---|
| BACKBONE (LLAMA-MOE) | | | |
| | # layers $L$ | 12 | 24 |
| | Hidden size $d$ | 768 | 1024 |
| | FFN hidden size | $4d = 3072$ | $4d = 4096$ |
| | Attention heads | 12 | 16 |
| | Query groups (GQA) | 4 | 4 |
| | Activation | SwiGLU | SwiGLU |
| | Sequence length / max positions | 1024 / 1024 | 1024 / 1024 |
| | Dropout (attention / hidden) | 0.0 / 0.0 | 0.0 / 0.0 |
| | Tied input/output embeddings | no | no |
| MOE CONFIGURATION | | | |
| | # experts $E$ | 8 | 8 |
| | Top-$k$ selection | 1 | 1 |
| | Granularity | 1 | 1 |
| | Load-balancing aux-loss coeff. $\alpha$ | 1e-2 | 1e-2 |
| | $m$ (GHA iteration) | 1 | 1 |
| | $\eta$ (GHA learning rate) | 2e-5 | 5e-5 |
| OPTIMIZATION | | | |
| | Hardware | $4 \times$ RTX 6000 Blackwell | $4 \times$ RTX 6000 Blackwell |
| | Optimizer | AdamW ($\beta_1$=0.9, $\beta_2$=0.999) | AdamW ($\beta_1$=0.9, $\beta_2$=0.999) |
| | Weight decay | 0.01 | 0.01 |
| | Global / micro batch size | 512 / 64 | 512 / 64 |
| | Base learning rate | 4e-4 | 4e-4 |
| | Min learning rate | 5e-5 | 5e-5 |
| | LR schedule | cosine decay | cosine decay |
| | Gradient clipping | 1.0 | 1.0 |
| | Precision | bfloat16 | bfloat16 |
| | Training iterations | 60,000 | 60,000 |
| | Total training tokens | 30B | 30B |

# F. Full Experimental Results on Synthetic-Data

## F.1. Evolution of Interpolation Coefficient $\alpha$

We provide the full results on the dynamics of the interpolation coefficient $\alpha$ across all expert sizes $K$ and top-$k$ settings, extending the representative trends shown in the main paper. For each configuration, we compute (i) the slope of the mean $\alpha$ over epochs and (ii) the temporal evolution of $\alpha$ throughout training. Specifically, for each epoch we log per-expert $\alpha$ values, average them across experts and random seeds, and then fit a linear regression of $\alpha$ against epochs to obtain the slope. This slope reflects the overall direction of $\alpha$'s change during training. In addition, we plot the mean $\alpha$ at every epoch with standard deviation across seeds, showing the full trajectory of its evolution.

Figure 8 confirms the consistent downward trend of $\alpha$, observed across all $K$ and $k$ configurations. This indicates that the gating mechanism increasingly shifts toward reliance on the GHA-driven subspace as training progresses, reducing dependence on the purely learnable gating matrix. While the absolute changes in $\alpha$ are small in magnitude, the uniformity of the trend across settings demonstrates the robustness of STAR's balance adaptation. These extended results reinforce our main finding that incremental subspace learning via GHA is progressively prioritized during training, leading to stable expert specialization.

*Table 7.* Detailed training hyper-parameters and configuration for GLUE fientuning experiments.

| Config | CoLA | MRPC | QNLI | MNLI | RTE |
|---|---|---|---|---|---|
| $m$ (GHA iteration) | 3 | 3 | 3 | 3 | 3 |
| $\eta$ (GHA learning rate) | 2e-5 | 2e-5 | 2e-5 | 2e-5 | 2e-5 |
| Epoch | 10 | 7 | 3 | 3 | 9 |
| Learning rate | 2e-5 | $\{2e\text{-}5, 3e\text{-}5\}^*$ | 2e-5 | 2e-5 | $\{2e\text{-}5, 3e\text{-}5\}^*$ |
| MoE layer | $\{10, 12\}^*$ | 10 | $\{10, 12\}^*$ | 10 | $\{10, 12\}^*$ |
| LR schedule | Linear | Linear | Linear | Linear | Linear |
| Weight decay | 0.0 | 0.0 | 0.0 | 0.0 | 0.0 |
| Train Batch size / GPU | 32 | 32 | 32 | 32 | 32 |
| Eval Batch size / GPU | 8 | 8 | 8 | 8 | 8 |
| GPU | | | $1 \times$ A6000 (48G) | | |

*Table 8.* Training and evaluation configuration for the vison OOD experiments on ViT-S/32. We follow the GMoE integration strategy (Li et al., 2023).

| Category | Hyperparameter | Value |
|---|---|---|
| BACKBONE AND DATA | | |
| | Backbone | ViT-S/32 |
| | Pretraining dataset | ImageNet-1k |
| | OOD evaluation dataset | ImageNet-C (15 corruptions, severities 1, 3, 5) |
| MOE CONFIGURATION | | |
| | # experts $E$ | 6 |
| | Top-$k$ routing | $k = 2$ |
| | $m$ (GHA iteration) | 3 |
| | $\eta$ (GHA learning rate) | 2e-5 |
| OPTIMIZATION | | |
| | GPU | $1 \times$ RTX A6000 |
| | Learning rate | 5e-5 |
| | Weight decay | 0.0 |
| | Dropout | 0.1 |
| | Batch size | 256 |
| | Training steps | 10,000 |

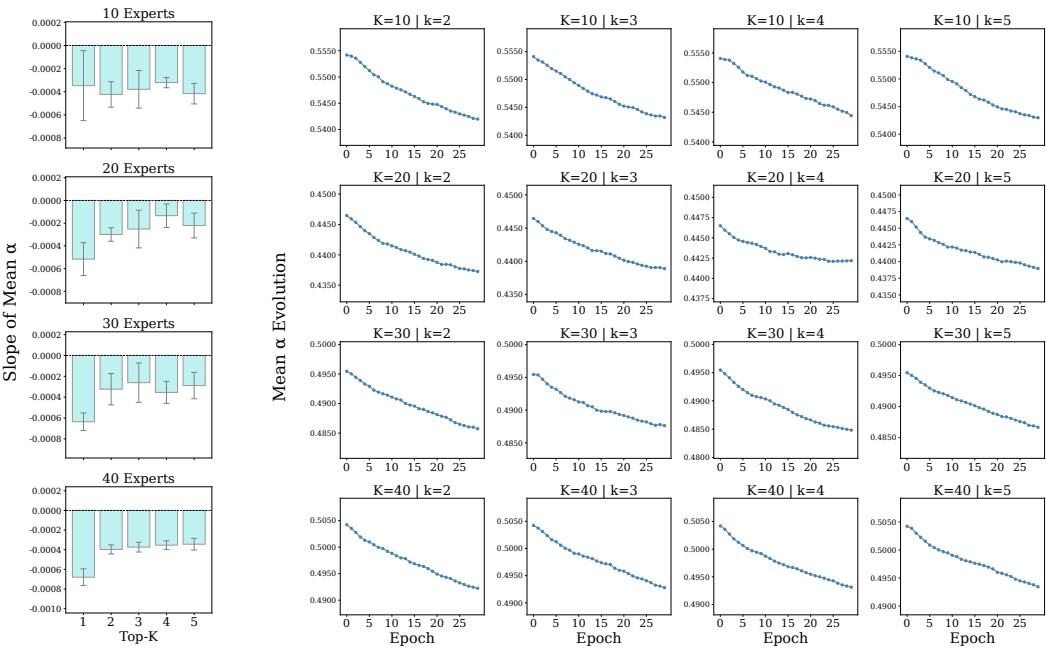

*Figure 8.* **Evolution of Interpolation Coefficient $\alpha$.** Left: Average slope of the mean $\alpha$ across different Top-$k$ settings. Right: Temporal evolution of mean $\alpha$ throughout training epochs.

