# OpenReview forum: "STAR: Rethinking MoE Routing as Structure-Aware Subspace Learning"
_ICML.cc/2026/Conference — ICML 2026 regular_

### Official Review · Reviewer_CTez · 2026-03-04

**Soundness:** 3
**Presentation:** 3
**Significance:** 2
**Originality:** 3
**Overall Recommendation:** 4
**Confidence:** 3

**Summary:**

This paper aims to improve the expert routing of MoE by formulating MoE routing as a subspace learning problem rather than using simple linear projection with limited awareness of input representation. To do so, the method augments standard learnable routing with an evolving principal subspace that tracks dominant input structure, and aligns routing decisions with input structure, along with the task-supervision from the learnable gate, so that more balanced expert specialization is achieved.

**Compliance With Llm Reviewing Policy:**

Affirmed.

**Key Questions For Authors:**

What are limitations of STAR, say, in what situation it fail to improve performance?

**Limitations:**

Limitations are not discussed in the paper.

**Strengths And Weaknesses:**

Strengths

-The proposed method that applies subspace learning for addressing the routing of MoE is novel. Rather than force load balancing as a consequence by simple regularization, it is more data-aware to drive the consequence of load balance.

-theoretical analyses are conducted to support claims.

-comprehensive experiments for evaluating the method. Also, computational cost is analytically and empirically measured.

-The writing and overall structure are clear to read.

Weaknesses:

-Improvement is not significant in general, and some tables do not have standard deviations, so not sure whether the improvement is statistically significant.

-The limitations are not discussed. In what situations they fail to improve, it is good to know when to apply the method.

---

> ### Author Rebuttal · Authors · 2026-03-31
>
> We sincerely thank reviewer CTez for careful reading and raising the constructive and valuable feedbacks, which help us sharpen both the presentation and the technical positioning of our work.
>
> ### **1. Robustness across seeds**
>
> - We add a multi-seed run table for the zero-shot commonsense evaluation on  LLaMA-MoE (182M), reporting mean and standard deviation across tasks. Due to limited time, we currently provide multi-run statistics for STAR on pretraining setup first, but we will update the remaining standard deviations for other experiments.
>
>     #### **Table 14. Mutli-seed run results for commonsense reasoning on LLaMA-MoE (182M)**
>
>     | Model (3 Runs) | ARC-c | ARC-e | BoolQ | HellaSwag | LAMBADA |  PIQA |  RACE |   Avg |
>     | -------------- | ----: | ----: | ----: | --------: | ------: | ----: | ----: | ----: |
>     | STAR (Avg.)    | 23.21 | 44.46 | 57.45 |     32.82 |   34.20 | 64.11 | 28.55 | 40.68 |
>     | (Std.)         |  2.73 |  3.82 |  3.00 |      2.14 |    0.83 |  0.55 |  0.34 |  0.37 |
>
>
> ### **2. Limitations**
>
> - In the architectural setup where the expert size closes to the model dimension, i.e., $K \approx d, d'$, the complexity caused by GHA updates can be less negligible. Practically, $K$ affects the routing cost of STAR linearly through $O(BKd)$ to generate routing logits, and  additionally introduces a basis-mixing term $O(K^2 d)$. Under the regime where $K \approx \min(d,d')$, this overhead becomes comparable to $O(Bkdd')$, which is the dominant complexity from per-expert FFN computation under top-$k$ setting.
>
>     - However, this setup $K \approx \min(d,d')$ is highly unlikely considering the practical real-world MoE setting where $K \ll \min(d,d')$. Typically, the number of experts is only a small fraction of the hidden dimension around 0.1\%-3\%, so the additional STAR overhead remains well below the dominant per-expert computation in practical setup.
>
>         #### **Table 16. Configurations for recent large-scale MoE frameworks**
>
>     - | Model  | $d_{\text{model}}$ | Experts per layer ($K_{\text{total}}$) | Active experts | $K_{\text{total}} / d_{\text{model}}$ |
>         | --------------------- | ------- | ------------------------------- | ------------------------- | ----------------- |
>         | DeepSeekMoE-16.4B [1] | 2048    | 64 routed + 2 shared  | top-6 routed + all shared | **3.2%**          |
>         | Qwen1.5-MoE-A2.7B [2] | 2048    | 60 experts                  | top-4                     | **2.9%**          |
>         | LLaMA-MoE-v1-3.5B (2/8) [3] | 4096    | 8 experts                   | top-2                     | **0.2%**          |
>         | Mixtral-8×22B [4]  | 6144    | 8 experts                   | top-2                     | **0.13%**         |
>
>
>
>
> ------------------------------------------
>
> [1] Dai, Damai, et al. "Deepseekmoe: Towards ultimate expert specialization in mixture-of-experts language models." arXiv preprint arXiv:2401.06066 (2024).
>
> [2] Ahmed, Imtiaz, et al. "Qwen 2.5: A comprehensive review of the leading resource-efficient llm with potentioal to surpass all competitors." Authorea Preprints (2025).
>
> [3] Zhu, Tong, et al. "Llama-moe: Building mixture-of-experts from llama with continual pre-training." arXiv preprint arXiv:2406.16554 (2024).
>
> [4]-1 Jiang, Albert Q., et al. "Mixtral of experts." arXiv preprint arXiv:2401.04088 (2024).

---

> > ### Author Rebuttal · Reviewer_CTez · 2026-04-03
> >
> > My concerns have been addressed.

---

> > > ### Author Response · Authors · 2026-04-03
> > >
> > > We are glad that our rebuttal addressed all your concerns. Thank you for your careful reading and valuable feedbacks.

---

### Official Review · Reviewer_jMeQ · 2026-03-10

**Soundness:** 2
**Presentation:** 2
**Significance:** 2
**Originality:** 2
**Overall Recommendation:** 4
**Confidence:** 4

**Summary:**

This paper proposes STAR, that augments standard linear gating with an unsupervised PCA-based branch learned via the GHA, combined with a learnable mixing matrix R and interpolation coefficients. The experimental coverage is comprehensive, spanning synthetic tasks, language pretraining and finetuning, vision OOD benchmarks, and test-time adaptation.

**Compliance With Llm Reviewing Policy:**

Affirmed.

**Final Justification:**

I thank the authors for the detailed rebuttal and the additional ablation results, which are helpful. However, my main concern remains only partially resolved, as I still find some tension between the paper's original motivation around structure-awareness and the analysis of what drives the balancing effect. Overall, the rebuttal improves my assessment but does not fully remove my concerns, so I raise my score from 3 to 4. I encourage the authors to refine the wording of the motivation and contribution claims in a revised version so that they more precisely match the evidence provided.

**Key Questions For Authors:**

The number of principal components is set equal to the number of experts K. Have you experimented with decoupling these two quantities, and if so, how sensitive is performance to this choice?

**Limitations:**

The paper has discussed limitation.

**Strengths And Weaknesses:**

Strength:
1. Improving MoE routing beyond simple linear gating is an important problem, and the paper explores an interesting angle by incorporating unsupervised data-driven signals into routing.
2. The experimental coverage is comprehensive, spanning synthetic tasks, language pretraining, vision fine-tuning, and out-of-distribution scenarios across multiple domains and settings.

Weakness:
1. The motivation is incoherent. The paper argues that conventional linear gating lacks awareness of input structure, which leads to imbalanced routing, and that making routing structure-aware resolves this. But no logical or theoretical justification is provided for why awareness of input structure should lead to balanced routing. In fact, the paper's own analysis proves the opposite: Proposition 4.2 shows that routing along principal components directly inherits the eigenvalue hierarchy, producing maximally imbalanced expert utilization. Balance instead arises from the mixing matrix R (Proposition 4.3), a mechanism orthogonal to structure-awareness.
2. The core concept of input structure is never formally defined. The paper implicitly equates it with top principal components, but variance-maximizing directions are not necessarily the most relevant directions for routing decisions, a well-known distinction analogous to PCA vs. LDA. Without a precise definition, the claim that routing should be structure-aware is difficult to evaluate.
2. The ablation study does not include the most diagnostic baseline, which would be removing the GHA component entirely and adding the learnable mixing matrix R directly to the vanilla linear gate. Since R is shown to be the primary driver of load balance (Propositions 4.2 and 4.3), this missing comparison makes it difficult to isolate how much of the improvement comes from the PCA subspace versus from the additional learnable parameters alone.

---

> ### Author Rebuttal · Authors · 2026-03-31
>
> We sincerely thank reviewer jMeQ for raising the constructive and valuable feedbacks, which help us sharpen both the presentation and the technical positioning of our work.
>
> ### **1. Motivation clarification**
>
> - We appreicate the reviewer for the careful reading and we agree that our original motivation can be clarified more precisely. Our claim is not that principal component alignment by itself guarantees balanced routing. In fact, our analysis shows the opposite, without mixing ($R=I$), routing energy inherits the variance hierarchy and becomes imbalanced.
>
> - In STAR, the role of the GHA basis is to provide a data-dependent coordinate system that captures dominant input variation, while the role of $R$ is to decouple expert selection from this hierarchical ordering by input variance, and the interpolation with the learnable gate preserves direct task supervision with gradient-based optimization. We will make this decomposition explicit and avoid confusion that structure-awareness alone is sufficient for load balance in revised version.
>
> ### **2. Formal definition of input structure**
>
> - We thank the reviewer for pointing out that the term input structure should be defined formally.
>
> - **Formal definition**
>     - Let $x \in \mathbb{R}^d$ be a hidden representation with zero-mean $\mu := \mathbb{E}[x] = 0$. Then
>       $$
>       \Sigma_x := \mathrm{Cov}(x)=\mathbb{E}\big[xx^\top\big]
>       $$
>       denotes its covariance matrix. For a target rank $K$, we define the input structure of $x$ as the rank-$K$ principal subspace that minimizes the reconstruction error
>       $$
>       S_K^*=\arg\min_{U\in\mathbb{R}^{d\times K},U^\top U=I}
>       \mathbb{E}\big[\|x-UU^\top x\|_2^2\big].
>       $$
>     - Equivalently, $S_K^*$ is the $K$-dimensional subspace that maximizes the projected variance of the centered hidden representation.
>
> - Under this definition, GHA serves as an online estimator of the principal subspace. We emphasize that we do not claim this subspace alone is universally sufficient for routing. Rather, it provides an unsupervised structural signal that can be combined with the task-supervised learnable gate.
>
>
> ### **3. Extended ablation on $R$**
>
> - We thank the reviewer for this suggestion. To remove the possibility that the gain comes merely from the additional learnable matrix $R$, we add a baseline that removes the GHA branch and instead applies to the standard linear gate, i.e.,
>     - $$l_{\mathrm{linear}} = W_g x, \quad l_{R} = R\, l_{\mathrm{linear}}.$$
>     - We note that this control is algebraically equivalent to a reparameterized linear gate, but it is still a useful empirical check for whether the improvement can be attributed primarily to the additional learnable matrix rather than the learned structure-aware subspace.
>
>     - #### **Table 15. Extended ablation results**
>
>         | Algorithms (K, k)              | CoLA          | MRPC          | QNLI          | MNLI          | RTE            | Average |
>         |-------------------------------|---------------|---------------|---------------|---------------|----------------|---------|
>         | STAR (8,4)  | 66.62 ± 0.65  | 89.68 ± 0.20  | 92.61 ± 0.10  | 86.74 ± 0.11  | 75.57 ± 0.68   | 82.24 |
>         | STAR w/o R (8,4)   | 64.57 ± 0.82  | 90.12 ± 0.50  | 92.54 ± 0.04  | 86.38 ± 0.06  | 72.56 ± 1.77   | 81.23 |
>         | STAR w/o Interpolation (8,4) | 66.02 ± 1.32  | 89.56 ± 0.31  | 92.39 ± 0.17  | 86.51 ± 0.12  | 73.53 ± 2.45   | 81.60 |
>         | No GHA + R on linear gate (8,4) | 64.45 ± 1.53  | 90.00 ± 0.05  | 92.55  ± 0.29  | 86.39 ± 0.08 |  74.85 ± 1.62 | 81.65 |
>
> - Our results show that applying $R$ directly to linear gate doesn't recover the full gain of STAR, supporting that the benefit does not come from the extra parameters alone and from the input-awareness with GHA-driven basis.
>
> - Regarding the reviewer’s question on decoupling K, we've not yet explored this design choice. In the current study, we set the subspace rank equal to the number of experts for clear interpretation. We agree that decoupling these two quantities can be an important direction, since a lower rank may provide a more compact and effective structural summary. We will try to include this as a limitation and valuable future work to be done.

---

> > ### Author Rebuttal · Reviewer_jMeQ · 2026-04-03
> >
> > I thank the authors for the thorough response and the effort to provide new ablation results.
> >
> > The extended results in Table 15 are informative, but they also suggest that R is the primary driver of improvement (1.0), while GHA contributes modestly (0.6). This partially confirms rather than resolves my original concern. I will keep my score.

---

> > > ### Author Response · Authors · 2026-04-03
> > >
> > > Thank you for the follow-up. We respectfully believe that Table 15 should not be interpreted as an additive decomposition such as "$R$ contributes 1.0, while GHA contributes 0.6", because STAR is not a sum of independent modules. Its router is defined jointly by the learned subspace $V$, the mixed subspace $Z = RV$. As a result, the contribution of each part is interaction-dependent, rather than cleanly separable.
> > >
> > > In fact, our analysis already clarifies the distinct but complementary roles of the two components. The role of $R$ is to remove the variance-ordering bias of direct principal-component routing, while the GHA branch provides the data-aligned coordinate system on which this mixing operates. Thus, $R$ is important, but it is not meaningful in isolation from the learned structure it mixes.
> > >
> > > Table 15 should be read as evidence that $R$ is the mechanism that makes structure-aware routing usable, not as evidence that structure-awareness is secondary. **Without GHA, $R$ only remixes an unconstrained linear gate and without $R$, GHA remains tied to the variance hierarchy. STAR improves because these two components work together**. GHA provides meaningful structure, and $R$ converts that structure into balanced and effective expert selection. In this sense, **STAR is a coupled routing design, not a set of independent sum of gains that can be cleanly factorized item by item**.

---

### Official Review · Reviewer_c9BR · 2026-03-10

**Soundness:** 3
**Presentation:** 3
**Significance:** 2
**Originality:** 3
**Overall Recommendation:** 4
**Confidence:** 3

**Summary:**

In this paper, the authors propose STAR, a new MoE routing learning method. In STAR, given an input token embedding, the router learns the top-K principal components of the data distribution through GHA, along with a mixing matrix R to augment the vanilla router learning. A gate is used to balance the contribution between STAR and the vanilla router. The authors evaluate the method on synthetic datasets to show that STAR achieves better loss than the baseline across different top-K settings and better load balance even without a load balancing loss. Next, the authors also adapt STAR to large-scale MoE pretraining and fine-tuning, showing its effectiveness.

**Compliance With Llm Reviewing Policy:**

Affirmed.

**Final Justification:**

After the rebuttal, the authors provide extensive additional experimental results, which solve most of my concerns. I increased my score to 4 accordingly.

**Key Questions For Authors:**

See above, and please clarify with me if I misunderstood any part of the paper or the setting.

**Limitations:**

1. the cost of the proposed method on training and inference is questionable if we scale the model to more product setting (much larger k and K), and also unclear how well the method perform on balancing different expert in a single batch in more challenge setting.

**Strengths And Weaknesses:**

## Strengths

1. The paper is clearly written and easy to follow.

2. The idea of learning principal components from input token embeddings to guide the routing is both interesting and theoretically sound.

3. The authors provide in-depth analysis of their method.

## Weaknessness

1. The major weakness, in my view, is the gap between the synthetic evaluation and real-world large-scale training. Specifically, STAR achieves great results in synthetic evaluations across different settings, and better load balance even without auxiliary loss, which is quite impressive. However, the generation of synthetic data following HHM is relatively simple and makes it easy for the model to learn the correct components and achieve good generalization on the test set (which follows the same rule). In contrast, real-world training is much more complicated. Please see the concerns below.
2. The authors did not report the pretraining loss comparison between STAR and different baselines, which is a more direct metric for evaluating the effectiveness of the proposed method than downstream tasks. Meanwhile, in the downstream task evaluation, STAR does not show a clear advantage compared to the baselines.
3. In the real-world setting, the authors only set top-1 routing from 8 experts, which is quite restricted and does not align with current large-scale MoE models (which typically use much more sparse and fine-grained experts). I am wondering how STAR would perform if we increase both top-k and the total number of experts.
4. The authors did not show the load balance comparison in the real-world setting (including both pretraining and fine-tuning), which leaves the question of whether STAR can maintain its balancing ability under more complex scenarios.
5. Although the authors provide complexity analysis, could the authors clarify the experimental setting (e.g., what are the top-k and total number of experts K)? How would varying k and K affect the practical cost in both training and inference?
6. The authors mention that in the real-world setting, the iteration m is set to 1–3. I am wondering whether this is sufficient for effective component learning, and how the authors determined this number.

---

> ### Author Rebuttal · Authors · 2026-03-31
>
> We sincerely thank reviewer c9BR for careful reading and raising the constructive and valuable feedbacks, which help us sharpen both the presentation and the technical positioning of our work.
>
> ### **1. Results on real experiments**
>
> - **Pretraining loss curves**
>
>     - [Figure 9. Training curves of different MoE models.](https://anonymous.4open.science/r/ICML-2026-Discussion-STAR-0670/Figure%209.pdf)
>     - Following the reviewer’s suggestion, we add loss curves on the pretraining setup. We compare STAR against two representative baselines, Vanilla+LB, and second-best ReMoE under the same training setup. As shown figure, STAR achieves a consistently lower training loss throughout training, supporting that its advantage persists at pretraining.
>     - While the absolute improvements are moderate, this is expected as the compared baselines are already tightly clustered with small margins between them. In such a setting, a consistent improvement over the best prior baseline can be meaningful. In Table 1, STAR outperforms the second-best method by 0.75 in average (40.78 vs. 40.03), suggesting that its benefit is non-trivial even against competitive large-scale MoE routers.
>
> - **Larger scale real-word experiment**
>
>     - We agree that the top-1/8 experts is relatively restricted considering the large-scale MoE setup. We chose this configuration primarily to follow the same pretraining setup given from ReMoE, so that the comparison remains fair.
>     - [Table 13. Performance comparison on GLUE. (K=16)](https://anonymous.4open.science/r/ICML-2026-Discussion-STAR-0670/Table%2013.pdf)
>         - To address this concern, we include a larger-scale experiment on GLUE with K=16 and multiple top-k choices. STAR remains superior across (16,1), (16,2), and (16,4), and achieves the best average result at STAR (16,4), 82.11, compared to 81.69 for Vanilla+LB (16,4) and 81.64 for DynMoE.
>
> - **Load balance under real-world setting**
>
>     - [Figure 10. Routing Specialization and Load Balance.](https://anonymous.4open.science/r/ICML-2026-Discussion-STAR-0670/Figure%2010.pdf)
>     - We agree with the reviewer that real-world load-balance analysis is important. To address this, we add a load balance analysis on GLUE finetuning (Figure 8). STAR shows the largest increase in inter-expert distance from early to late training (+2.0%), suggesting stronger specialization.
>     - At the same time, the Lorenz curves show that STAR achieves the most balanced Top-1 token assignment, and remains competitive under Top-4 activations, staying close to Vanilla+LB while clearly improving over Vanilla and STAR w/o $R$. These results support that the balancing behavior of STAR is preserved in a more realistic fine-tuning scenario, even without auxiliary load-balancing regularization.
>
> ### **2. Complexity analysis**
>
> - For complexity analysis, we use $(K, k)=(8,2)$ setup. In general, increasing top-$k$ affects the dominant per expert FFN computation already present in standard MoE, taking $O(Bkdd')$ where $d$, $d'$ are hidden dimensions. Increasing $K$ affects the routing cost of STAR linearly through $O(BKd)$ with respect to $m$, and additionally a basis-mixing term $O(K^2 d)$. Under the practical regime we consider, where $K \ll d,d'$ and $m$ is small, this overhead is negligible relative to $O(Bkdd')$.
>
>     - $K \ll \min(d,d')$ is not an arbitrary assumption, but typical real-world scenarios in recent MoE frameworks. The number of experts is only a small fraction of the hidden dimension, so the additional STAR overhead remains well below the dominant expert computation.
>
>         #### **Table 16. Configurations for recent large-scale MoE frameworks**
>
>         | Model  | $d_{\text{model}}$ | Experts per layer ($K_{\text{total}}$) | Active experts | $K_{\text{total}} / d_{\text{model}}$ |
>         | --| - | -- | - | --|
>         | DeepSeekMoE-16.4B [1] | 2048 | 64 routed + 2 shared  | top-6 routed + all shared | **3.2%**        |
>         | LLaMA-MoE-v1-3.5B (2/8) [2] | 4096    | 8 experts  | top-2 | **0.2%** |
>         | Mixtral-8×22B [3]  | 6144 | 8 expert | top-2 | **0.13%** |
>
> ### **3. Iteration number $m$**
> - We use small $m$ because the basis-quality analysis in Appendix C. Figure 7 shows that even a few GHA iterations ($m \le 3$) already recover the dominant structure well, while larger $m$ progressively improves the approximation toward full-batch SVD.
> - In our GLUE ablation in Table 6 (Appendix D), the $(8,2)$ setting achieves average scores of 82.10 ($m=1$), 81.91 ($m=2$), and 82.11 ($m=3$), showing even $m \in [1, 3]$ is sufficient in practice.
>
> ---
> [1] Dai, Damai, et al. "Deepseekmoe: Towards ultimate expert specialization in mixture-of-experts language models." arXiv preprint arXiv:2401.06066 (2024).
>
> [2] Zhu, Tong, et al. "Llama-moe: Building mixture-of-experts from llama with continual pre-training." arXiv preprint arXiv:2406.16554 (2024).
>
> [3]-1 Jiang, Albert Q., et al. "Mixtral of experts." arXiv preprint arXiv:2401.04088 (2024).

---

> > ### Author Rebuttal · Reviewer_c9BR · 2026-04-03
> >
> > I really appreciate the effort the authors made to address my concerns. I have a few follow-up questions:
> >
> > 1. The pretraining loss curve is a great add-on. But I am wondering whether the authors could further provide an evaluation loss/PPL comparison after pretraining on a small held-out corpus? I would like to know the exact number of the improvement.
> >
> > 2. I thank the authors for the additional load balance comparison in a real-world setting. Could the authors provide an explanation for the increase in k=1 to k=4 hurting the performance of STAR on load-balance compared to vanilla+LB (Figure 10 right)? Meanwhile, is this the model you used in Table 13? What makes the STAR achieve an average better performance than the vanilla+LB with k=4 when the load balance is not as good as the vanilla+LB?
> >
> > 3. I appreciate the additional complexity analysis, and it makes sense. But I am curious about the practical cost of STAR when changing the k and K (an extension to the current Table 5).

---

> > > ### Author Response · Authors · 2026-04-03
> > >
> > > We appreciate the reviewer for carefully reading our rebuttal and further follow-up questions.
> > >
> > > ### **1. Pretrainng Loss/Perplexity**
> > > We evaluate the final pretrained checkpoints on a 10M held-out validation subset of Pile.
> > > - LLaMA-MoE (182M) pretrained on 30B Pile tokens
> > > - Evalution on 10M (10,485,760) Pile tokens
> > >
> > > | Model | Validation Loss | Validation PPL |
> > > |------|---------------:|---------------:|
> > > | Vanilla + LB | 1.99 | 7.32 |
> > > | ReMoE | 1.94 | 6.95 |
> > > | **STAR** | 1.90 | 6.64 |
> > >
> > > STAR achieves lower held-out validation loss and perplexity in comparison to other baselines.
> > >
> > > ### **2. Load-Balance and Performance**
> > >
> > > Thank you for this helpful follow-up. We first note that load-balance analysis in Figure 10 (right) was checked on the same GLUE setting as Table 3. We agree that in the top-4 setting, expert usage in STAR is less balanced than Vanilla+LB. Although load balance is a useful diagnostic, however, we do not expect load balance always maps perfectly to downstream performance. Balancing should be a consequence of effective routing derived from better specialization rather than the primary target itself. Explicitly forcing uniform usage can over-constrain routing and introduce a interfering bias that is not necessarily aligned with the downstream task.
> > >
> > > We believe this is what happens in the top-4 case. STAR is designed to make expert selection more input-relevant, not to explicitly enforce uniformity over the entire top-4 co-activation pattern. By design, top-1 and top-4 measure different things. In top-1, only the primary expert matters, so the routing can remain fairly balanced. In top-4, the metric is mainly affected by co-activation patterns where the experts are repeatedly selected together. STAR may reuse a similar subset of experts for similar inputs because those experts are genuinely relevant. However, this co-selection is different from load collapse, but rather collaboration. This is why STAR can look less balanced in the top-4 setting while still achieving better task performance.
> > >
> > >
> > > ### **3. Complexity Analysis on varying $(K, k)$**
> > >
> > > - We apologize for the earlier mistake, the complexity results in Table 5 correspond to the *(8,4) setting, not (8,2).
> > > - As requested, we add Table 15 as an extension of Table 5, which reports the practical runtime and peak memory of STAR under varying $(K,k)$ settings. The measured trend is consistent with our analysis. Increasing $k$ mainly raises the shared expert-computation cost, while increasing $K$ adds some routing overhead for STAR, but the overall increase remains modest.
> > >
> > >     - [Table 15. Extended complexity analysis on varying MoE setup](https://anonymous.4open.science/r/ICML-2026-Discussion-STAR-0670/Table%2015.pdf)

---

### Decision · Program_Chairs · 2026-04-30

**Decision:**

Accept (regular)

**Comment:**

The paper received positive reviews, with all three reviewers (c9BR, jMeQ, CTez) recommending weak accept. The AC has read the reviews, author responses, and discussions. After rebuttal, the reviewers acknowledge their concerns have been addressed through extensive new results (pre-training loss/perplexity, load-balance analysis, larger-scale real-world experiments, multi-seed runs, etc). The AC concurs that the structure-aware subspace formulation for MoE routing is interesting, and the extensive experiments strengthen the contribution. Therefore, the AC decides to accept the work. The authors are encouraged to incorporate the promised clarifications on motivation wording and any remaining minor points in the final version.